# X-ray study of ferroic octupole order producing anomalous Hall effect

Motoi Kimata [1]✉, Norimasa Sasabe[2], Kensuke Kurita[3], Yuichi Yamasaki[4,5,6,7], Chihiro Tabata[8], Yuichi Yokoyama [2], Yoshinori Kotani [2], Muhammad Ikhlas[9], Takahiro Tomita [9], Kenta Amemiya [7], Hiroyuki Nojiri[1], Satoru Nakatsuji[9,10,11,12], Takashi Koretsune [3], Hironori Nakao [7], Taka-hisa Arima [5,13] & Tetsuya Nakamura [1,2,14]

Recently found anomalous Hall, Nernst, magnetooptical Kerr, and spin Hall effects in the antiferromagnets $Mn_3X$ (X = Sn, Ge) are attracting much attention for spintronics and energy harvesting. Since these materials are antiferromagnets, the origin of these functionalities is expected to be different from that of conventional ferromagnets. Here, we report the observation of ferroic order of magnetic octupole in $Mn_3Sn$ by X-ray magnetic circular dichroism, which is only predicted theoretically so far. The observed signals are clearly decoupled with the behaviors of uniform magnetization, indicating that the present X-ray magnetic circular dichroism is not arising from the conventional magnetization. We have found that the appearance of this anomalous signal coincides with the time reversal symmetry broken cluster magnetic octupole order. Our study demonstrates that the exotic material functionalities are closely related to the multipole order, which can produce unconventional cross correlation functionalities.

[1] Institute for Materials Research, Tohoku University, Sendai, Miyagi 980-8577, Japan. [2] Japan Synchrotron Radiation Research Institute (JASRI), 1-1-1 Kouto, Sayo, Hyogo 679-5198, Japan. [3] Department of Physics, Tohoku University, Sendai, Miyagi 980-8578, Japan. [4] Research and Services Division of Materials Data and Integrated System (MaDIS), National Institute for Materials Science (NIMS), Tsukuba, Ibaraki 305-0044, Japan. [5] Center for Emergent Matter Science (CEMS), RIKEN, Wako 351-0198, Japan. [6] PRESTO, Japan Science and Technology Agency (JST), Tokyo 102-0076, Japan. [7] Institute of Materials Structure Science, High Energy Accelerator Research Organization, Tsukuba, Ibaraki 305-0801, Japan. [8] Institute for Integrated Radiation and Nuclear Science, Kyoto University, Kumatori, Osaka 590-0494, Japan. [9] Institute for Solid State Physics, University of Tokyo, Kashiwa, Chiba 277-8581, Japan. [10] Department of Physics, University of Tokyo, Hongo, Tokyo 113-0033, Japan. [11] The Institute for Quantum Matter, Johns Hopkins University, Baltimore, MD 21218, USA. [12] Trans-scale Quantum Science Institute, University of Tokyo, Hongo, Tokyo 113-8654, Japan. [13] Department of Advanced Materials Science, University of Tokyo, Kashiwa 277-8561, Japan. [14] International Center for Synchrotron Radiation Innovation Smart, Tohoku University, Sendai, Miyagi 980-8577, Japan. ✉email: motoi.kimata.b4@tohoku.ac.jp

The coupling between circularly polarized light and magnetic moments has various importance in condensed matter physics since this coupling is a source to induce a spin polarization and detection through the angular momentum transfer. Especially in the soft X-ray region, the photon energies correspond to the absorption edges from $2p$ to $3d$ states ($L_{2,3}$ edges) for $3d$ transition metal elements, and thus the energy-resolved spectroscopy is an efficient probe for microscopic properties of spin-split density of states (DOSs), such as element-specified magnetization, separation of spin and orbital magnetic moments. This technique is established as X-ray magnetic circular dichroism (XMCD), and is mainly applied to investigate electronic states in ferromagnetic materials so far[1–3]. On the other hand, for antiferromagnetic (AF) materials, in which the magnetization is mostly zero due to their compensated magnetic structures, the XMCD is generally absent since their DOSs are spin-degenerated. In other words, the macroscopic breaking of time-reversal symmetry (TRS), which is typically driven by the presence of uniform magnetization, may be necessary to observe sizable XMCD signals.

Among many kinds of AF materials, the non-collinear antiferromagnets $Mn_3X$ with hexagonal (X = Sn, Ge, Ga)[4–7] and cubic (X = Pt)[8] structures are attracting much attention due to their large anomalous Hall effects (AHE). Especially in the hexagonal $Mn_3Sn$, various types of anomalous responses such as large anomalous Hall, Nernst, magnetooptical Kerr, and unconventional spin Hall effects have been established[9–12]. Although these extraordinary responses require breaking of TRS, the observed spontaneous magnetization is vanishingly small ($\sim 2$ m$\mu_B$/Mn) compared with that of typical ferromagnets since the sublattice magnetic moments are mostly compensated by the inverse triangular magnetic structure (Fig. 1a)[4,13–16]. Thus, to explain the sizable responses of these materials, another driving

force is expected. The origin is currently understood with an order parameter called cluster magnetic octupole, which in principle breaks TRS without net magnetization[17]. However, the presence of magnetic octupole order, has not been well established due to the lack of a suitable experimental approach to detect magnetic octupole moments.

Recently, a theoretical study suggests a possibility to detect the cluster magnetic octupole order by XMCD through the coupling between circular polarized X-ray and anisotropic magnetic dipole moment, which is known as the $T_z$ term[18]. Since both the magnetic dipole moment in XMCD and cluster magnetic octupole are arising from the spatial distribution of spin density, both of them can similarly be described as coupled operators of spin and quadratic moments[18–22]: the key point for the cluster multipole observation is the similarity of theoretical expression between the $T_z$ term and cluster magnetic octupole. Moreover, since the $T_z$ term is generally coming from the intra-atomic magnetic dipole moment, it is not detectable by the static magnetometry[23,24]. These discussions suggest the totally different XMCD properties, e.g., spectral shape and magnitude of the obtained moment, are expected in the multipole-mediated XMCD compared with that originating from the conventional spin moment XMCD, where only the spin-split DOS plays an essential role.

## Results and discussions

Figure 1a shows a schematic of the magnetic structure of $Mn_3Sn$ and the experimental configuration for XMCD. The right and left circularly polarized X-ray beams are irradiated to the cleaved (0001) surface of $Mn_3Sn$ single crystals with a tilted angle $\theta_{in}$. The angle between the magnetic field ($B$) and (0001) plane is defined as $\theta_B$. The X-ray energy corresponds to the transitions between $2p_{3/2}$ ($2p_{1/2}$) and $3d$ state called $L_3$ ($L_2$) absorption edges (Fig. 1b).

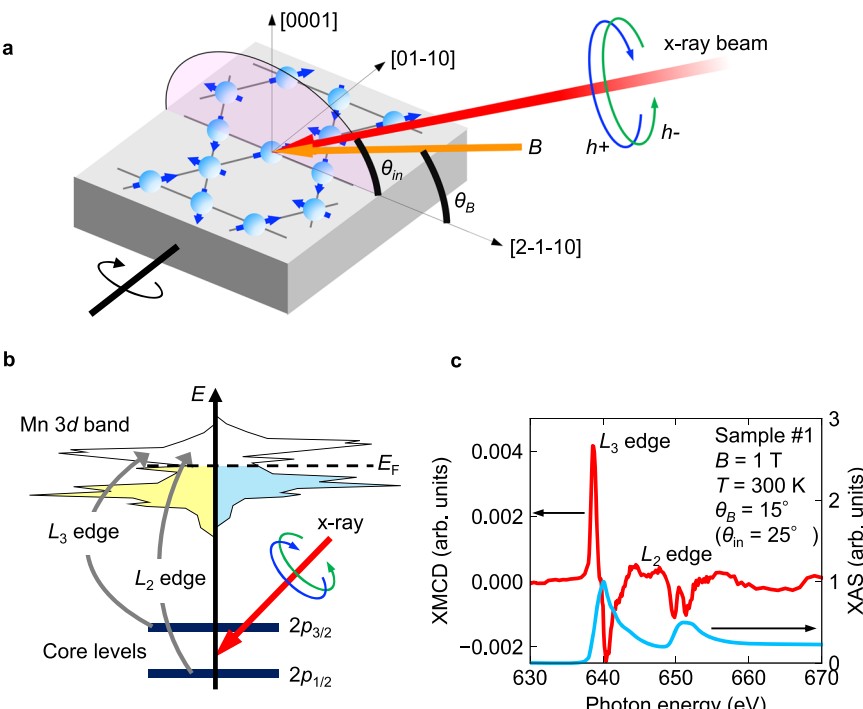

**Fig. 1 Experimental setup, principles for XMCD measurement, and typical XMCD spectrum in Mn₃Sn. a** Schematic of the magnetic structure of $Mn_3Sn$ and experimental setup. A circularly polarized X-ray beam was irradiated on the cleaved (0001) plane. $\theta_{in}$ ($\theta_B$) is an angle between (0001) plane and X-ray beam (magnetic field) within a plane containing [2–1–10] and [0001]. The angle between X-ray and magnetic field ($\theta_{in} - \theta_B$) is 10 degrees for samples #1 and #2, and $\theta_{in} - \theta_B = 0$ for sample #3. **b** Principle of XMCD. $L_3$ ($L_2$) edge corresponds to the excitation between $2p_{3/2}$ ($2p_{1/2}$) core level and unoccupied $d$ states. **c** Typical XMCD spectrum (left axis) and XAS (right axis) of $Mn_3Sn$ obtained for $B = 1$ T. The XMCD intensity is normalized by the peak intensity of XAS.

Figure 1c shows the typical XMCD (left axis) and X-ray absorption spectrum (XAS) (right axis) for $Mn_3Sn$. XMCD signal is obtained from the difference between the right and left circularly polarized X-ray absorptions. XAS shows a typical characteristic of metallic Mn compound[25,26], where broad absorption peaks with no clear multiplet structures are observed, indicating the formation of dispersive itinerant bands of Mn $d$ orbitals. As seen in Fig. 1c, clear positive and negative XMCD signals are observed for the $L_3$ and $L_2$ edges, respectively. This feature is indeed opposite to the case for the conventional ferromagnetic Mn compounds, where the sign of XMCD is negative (positive) for $L_3$ ($L_2$) edge[25,26]. The peak intensity of this XMCD signal is approximately 0.2–0.4% of the absorption. This value is small compared with that of typical $3d$ ferromagnetic metals (10–20% of XAS)[25,26]. However, as discussed below, the XMCD signal in $Mn_3Sn$ is still too large to be explained by the small spontaneous magnetization of $Mn_3Sn$, whereas the XMCD response of typical collinear antiferromagnet is almost absent[27].

Figure 2a shows the field orientation and strength (inset) dependence of the XMCD spectrum for sample #1. In the experiment for sample #1, the difference between $\theta_{in}$ and $\theta_B$ is always fixed at 10°. For $\theta_B = 15°$ ($\theta_{in} = 25°$), as shown in the inset, the spectral shape and intensity are essentially independent of the field strength, and a clear XMCD signal similar to that for $B = 1$ T is still observed even for $B = 0$ T, i.e., the remnant component. In contrast, for $\theta_B = 90°$ ($B||[0001]$, $B$ is perpendicular to the kagomé plane and $\theta_{in} = 100°$), the XMCD signal is almost zero for $B = 0$ T (sky-blue dashed line), and a small paramagnetic-like XMCD, i.e., the sign of XMCD is same with that of the ferromagnet, correspond to ~30 m$\mu_B$/f.u. (see the sum rule analysis

shown below) was observed for $B = 1$ T (blue solid line). This is reasonably consistent with the bulk magnetization by spin canting observed for $B||[0001]$. Note that the XMCD sign and spectral shape are totally different between $\theta_B = 15°$ and 90°. For example, the strongest peak position at the $L_3$ edge for $\theta_B = 15°$ (red arrow) is slightly lower than that for $\theta_B = 90°$ (blue arrow), and the shape of $L_2$ edge has double minimum for $\theta_B = 15°$, while that for $\theta_B = 90°$ shows a broad peak. To investigate the origin of this anomalous (opposite sign, field-independent, and lower energy peak) XMCD signal, we have measured $\theta_{in}$ dependence ($\theta_B$ is also changed together) of peak intensity at the $L_3$ edge (red arrow). As shown in Fig. 2b, the peak intensity follows $\cos\theta_{in}$ dependence with the incident angle $\theta_{in}$. Since the XMCD signal is proportional to the projection component parallel to the incident X-ray, this result indicates that the main contribution of unconventional XMCD originates from the magnetic moment lying on the kagomé plane. As shown by the red circles, this large anisotropy is also observed at 1 T, where the magnetizations for parallel and perpendicular directions to the kagomé plane are almost identical (Supplementary Fig. S1a). This fact also excludes the exchange-coupled impurity origin of present XMCD, where the signal sign and shape are expected to be always opposite to the magnetization. The lower energy minimum at the $L_3$ edge for $\theta_B = 90°$ (blue dashed arrow) is originating from the reduced in-plane XMCD due to the small in-plane component of incident X-ray with $\theta_{in} = 100°$ since $\cos 100° \approx -0.17$.

This highly anisotropic response of XMCD intensity is more clearly confirmed in the field-swept data at a constant photon energy. In this experiment, the photon energies for different field orientations were chosen at the peak of the XMCD signal for

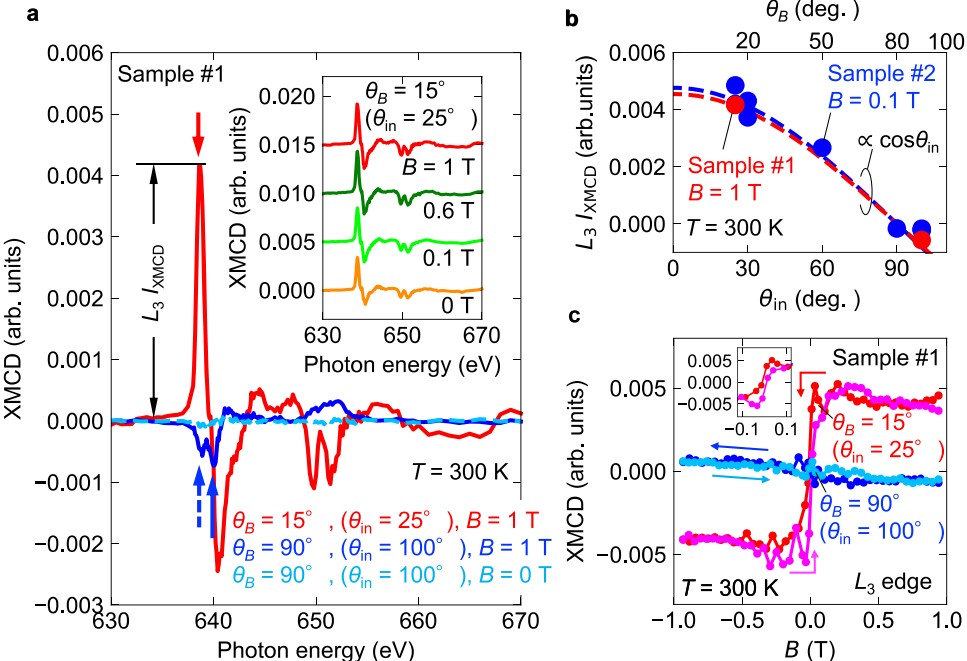

**Fig. 2 Field orientation and strength dependences of XMCD signals. a** Field-orientation dependence of XMCD spectrum for $B = 1$ T (red and blue curve), and XMCD for $\theta_B = 90°$ and $B = 0$ T (dashed sky-blue). For $\theta_B = 15°$ ($\theta_{in} = 25°$), the spectral shape is almost field independent, and the residual XMCD signal is still observed at 0 T as shown in the inset. To measure the XMCD signal at 0 T, 0.1 T is firstly applied, and the field is reduced to be 0 T. **b** $\theta_{in}$ dependence of XMCD peak intensity for $L_3$ edge ($\equiv L_3 I_{XMCD}$) and $B = 0.1$ T and 1 T. Corresponding field angle $\theta_B$ is also presented in the upper horizontal axis. The dashed line represents a fit proportional to $\cos\theta_{in}$ for the data. **c** Field-swept XMCD at constant photon energies of 638.8 eV and 640.0 eV for $\theta_B = 15°$ ($\theta_{in} = 25°$, red and magenta curves) and $\theta_B = 90°$ ($\theta_{in} = 100°$, blue and sky-blue curves), respectively. The photon energy for each field direction corresponds to the maximum XMCD intensity at the $L_3$ edge as indicated by red and blue solid arrows in **a**, respectively. Red (blue) and magenta (sky-blue) curves correspond to reducing and increasing field sweep processes, respectively. Field sweep directions are also shown by arrows with the same colors. A large ferroic response with a small switching field of ~0.01 T (see inset) is observed for $\theta_B = 15°$ ($\theta_{in} = 25°$), whereas a slight negative slope is observed for $\theta_B = 90°$ ($B||[0001]$ and $\theta_{in} = 100°$).

$\theta_B = 15°$ and 90° (red and blue arrows in Fig. 2a). In Fig. 2c, the large ferroic response is observed and the intensity mostly saturates above ~0.1 T for $\theta_B = 15°$, whereas the slight negative slope is observed for $\theta_B = 90°$. The negative slope for $\theta_B = 90°$ is consistent with the paramagnetic spectrum is shown in Fig. 2a: the negative slope for field-swept data is originating from the paramagnetic contribution. Note that the slightly negative slope at high field region is also observed for $\theta_B = 15°$, meaning that the XMCD response for $\theta_B = 15°$ is a sum of ferroic component and paramagnetic contribution. Although the switching field observed in XMCD (~0.01 T) is smaller than that of static magnetization measurement (~0.04 T, see Supplementary Fig. S1a) with a distinct sample. The discrepancy between XMCD and magnetization might be originating from sample dependence with extrinsic origin. The variation of the switching field is indeed observed in previous reports[4,9,11]. Importantly, the large anisotropic XMCD intensity even at a high magnetic field (~1 T) cannot be explained by the magnetization since the magnetizations for in-plane and out-of-plane field directions are mostly identical. Thus, these unique features of in-plane XMCD presented here evidently show a distinct origin of present XMCD from the conventional XMCD, where the intensity should be proportional to the static magnetization.

To estimate magnetic moments quantitatively, we apply the XMCD sum rule analyses to the present results. In general, the magnetic moments from the XMCD sum rules for $L_3$ and $L_2$ edges are obtained from the energy integrals of XMCD and XAS[3,28]:

$$\frac{m_L}{n_h} = -\frac{4q}{3r} \tag{1}$$

and

$$\frac{m_{S_{eff}}}{n_h} = -\frac{(6p - 4q)C}{r} \tag{2}$$

Here, $n_h$ is the number of holes (~4 in the present case[17]) in the $3d$ orbitals at a Mn site. $p$ and $q$ are the energy integrated values of XMCD for $L_3$ edge and whole energy range, respectively. $r$ is the energy integrated value of XAS for the whole energy range. $C$ is the correction factor originating from the $j$–$j$ mixing effect which is typically ~1.5 for Mn[29,30]. The degree of circular polarization is also considered for the estimation of magnetic moments. Equation (1) expresses that the total integral of the XMCD spectrum corresponds to the orbital moment ($m_L$). The effective spin moment $m_{S_{eff}}$ in Eq. (2) contains contributions from spin and magnetic dipole terms expressed as $-(2\langle S_z \rangle + 7\langle T_z \rangle)\mu_B$, where $\mu_B$ is the Bohr magneton, $\langle S_z \rangle$ and $\langle T_z \rangle$ are the expectation values of spin and magnetic dipole operators for the $3d$ state, respectively. In usual bulk ferromagnets with high symmetry, $\langle T_z \rangle$ is negligibly smaller than $\langle S_z \rangle$. Firstly, we discuss the case for $\theta_B = 90°$ ($B||[0001]$). As shown by open circles in Fig. 3a, $m_{S_{eff}}$ shows almost linear dependence with field strength for $B||[0001]$. The value is about ~30 m$\mu_B$/f.u. at 1 T. As mentioned above, the XMCD spectrum for $\theta_B = 90°$ ($\theta_{in} = 100°$) contains reduced in-plane signal since the incident X-ray has a small in-plane component correspond to $\cos 100° \approx -0.17$. This contributes to the overestimation of magnetic moments for this angle. Since $m_{S_{eff}}$ for in-plane direction is ~−60 m$\mu_B$/f.u. on average, the overestimate for $\theta_B = 90°$ direction is estimated to be ~10 m$\mu_B$/f.u. Thus, the out-of-plane magnetic moment at 1 T is estimated to be ~20 m$\mu_B$/f.u. This value is reasonably consistent with the magnetization observed in static measurement[4] (see also Supplementary Fig. S1a). The tiny $m_L$ terms for both in-plane and out-of-plane directions indicate that the orbital moments are nearly quenched in this material, which are consistent with the first-principles calculation. These facts imply that the XMCD for $\theta_B =$

90° can accurately evaluate the bulk magnetic moment of this material.

On the other hand, for $\theta_B = 15°$ ($\theta_{in} = 25°$, i.e., $B$ is nearly aligned with the kagomé plane), the magnetic moments obtained from the sum rule analysis completely disagree with the static magnetization: the value of $m_{S_{eff}}$ (solid circles) is negative and nearly independent of the field strength. Relatively large errors in $m_{S_{eff}}$ for $\theta_B = 15°$ is due to the uncertainly of integrated values of XMCD coming from the oscillating behavior above ~655 eV, which is not observed for $\theta_B = 90°$. Since the static magnetization for $B||[2-1-10]$ and $B||[0001]$ is almost identical at 1 T (Supplementary Fig. S1a), the magnitudes of spin moment $\langle S_z \rangle$ for both the field directions are also expected to be similar to each other. Thus, the origin of negative and field-independent $m_{S_{eff}}$ for $\theta_B = 15°$ could not be ascribed to the spin contribution $\langle S_z \rangle$ since the sign of $\langle S_z \rangle$ should be always positive. On the other hand, the sign of dipole term $\langle T_z \rangle$ can be both positive and negative depending on the spin and orbital configurations. Thus, the observation of negative $m_{S_{eff}}$ allows us to conclude that the XMCD parallel to the kagomé plane arises from the magnetic dipole term, which is unobservable in a static magnetization measurement. Indeed, the negative dipole term, which corresponds to the positive spectral shape at the $L_3$ edge, is reproduced by our spectral model calculations together with the orbital configuration determined by the first-principles calculations as shown below.

The importance of $T_z$ term-mediated XMCD is often discussed in surface magnetism, where the symmetry reduction of surrounding ligands plays an essential role[31]. If the present XMCD would be hosted by the surface magnetization, the signal would be independent of the bulk properties such as AHE and spontaneous bulk magnetization. In the well-ordered Mn$_3$Sn single crystals, the triangle magnetic structure with negative spin chirality is formed below 430 K, and another magnetic transition around 270 K was reported[32]. In the low-temperature phase below 270 K, the AHE and small spontaneous magnetization vanishes; it indicates that the TRS is preserved[32] (see also Supplementary Fig. S1b). This means that the ferroic octupole order is only predicted between 270 K and 430 K. Figure 3b shows XMCD spectra at 300 and 200 K. The XMCD mostly vanishes at $T = 200$ K. This means that the XMCD signal detect the bulk properties (which is a good evidence to show the bulk origin of present XMCD) since the temperature dependence of XMCD would be decoupled from the bulk properties if the present XMCD is arising from the independent surface magnetism from the bulk. The low-temperature phase below ~270 K is considered to be characterized as an incommensurate spiral magnetic order along the [0001] direction[32,33].

Here, we compare the observed XMCD signal to a single-atom model calculation (see "Methods" section) to reveal the origin of experimental observations. From the first-principles band calculation, the number of electrons per Mn atom is estimated to be ~5.8[17], so the high-spin $d^6$ electronic state for Mn (Mn$^{+1}$) is assumed in the single-atom calculation. Each Mn is bonded to four neighboring Mn atoms in the kagomé layer. In this calculation, an electron of the minority spin is assumed to occupy the $x^2$–$y^2$-type orbit, which is determined from our first-principles calculations, with the directions shown in the inset of Fig. 4a. The spectral calculation is performed for three magnetic sublattices labeled A, B, and C as shown in the insets of Fig. 4a–c, and then the sum of three spectra is taken as the total XMCD spectrum. Figure 4a, b shows typical calculated XMCD spectra for triangle magnetic structure of this material, which is characterized by the negative spin chirality of triangle magnetic structure[4,13,14]. The difference between Fig. 4a, b is the presence of small in-plane

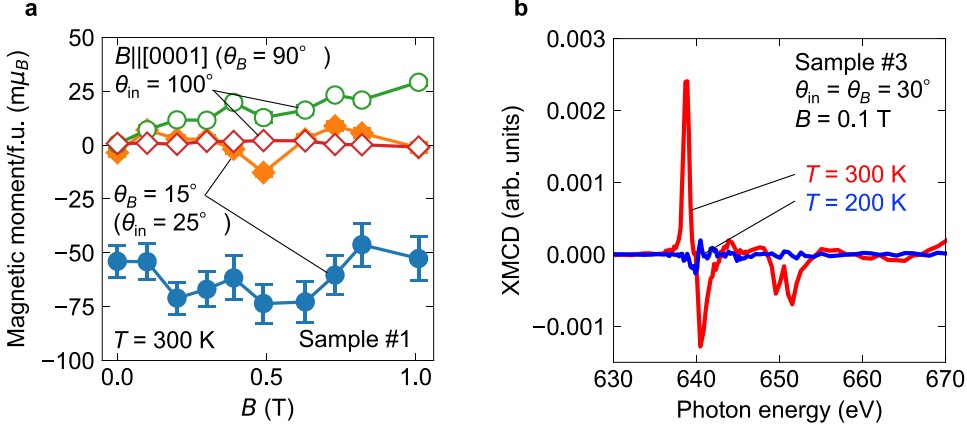

**Fig. 3 XMCD sum rule analysis and temperature dependence of XMCD spectra. a** Magnetic moments in $Mn_3Sn$ obtained from sum rule analyses for $\theta_B = 15°$ (solid symbols) and $B||[0001]$ (open symbols), respectively. Circles and diamonds correspond to the magnetic moments of $m_{S_{eff}}$ and $m_L$, respectively. Relatively large errors in $m_{S_{eff}}$ for $\theta_B = 15°$ are determined from the uncertainty of XMCD integration by the oscillating behavior above ~655 eV, which is observed only for $\theta_B = 15°$. **b** XMCD spectra for $T = 300$ K and 200 K. The XCMD intensity of Fig. 3b (sample #3) is almost 1.5–1.7 times smaller than that of Fig. 2a (sample #1). Although the origin of intensity variation is not clear, might be arising from extrinsic difference of sample conditions including surface contaminations and/or distinct sample batch.

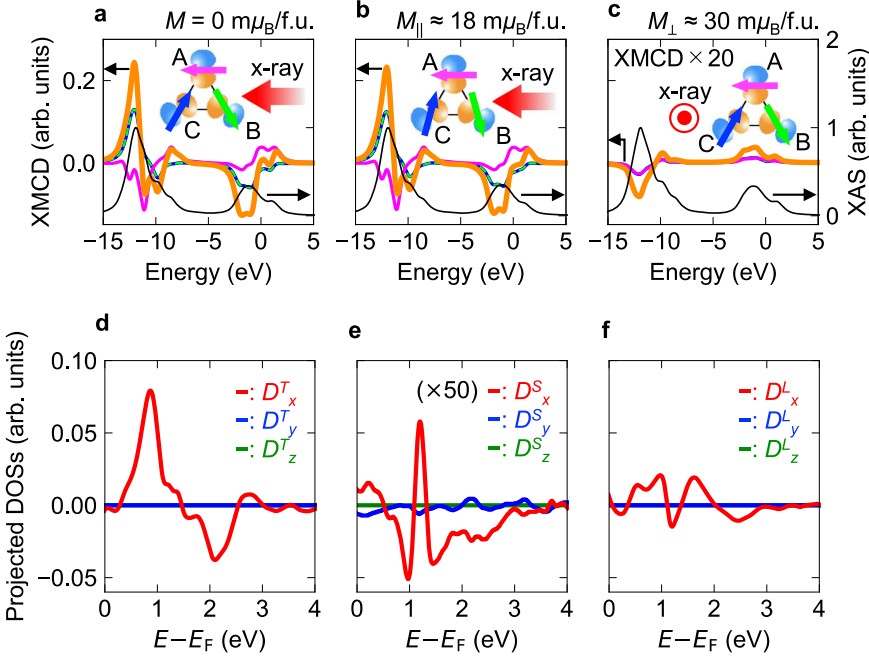

**Fig. 4 Calculated XMCD spectra and density of states. a–c** The XMCD and XAS spectra were obtained by the model calculation for different magnetic structures and X-ray configurations shown in the insets. For $Mn_3Sn$, $x^2–y^2$ type orbit is predicted to be the ground state for minority spins from our first-principles calculation as shown in the insets. The orbital arrangement obtained from the first-principles calculation is used in the atomic model calculation for XAS and XMCD. The individual XMCD response from each sublattice A, B, and C is indicated by magenta, dashed light green, and blue curves, respectively. The total sum of the XMCD spectrum is shown by the bold orange line. In **a**, the magnetic moments are completely canceled, while the magnetic moments on sublattice B and C in **b** are slightly tilted toward the X-ray direction to reproduce the situation with in-plane external field of ~1 T (the tilted angles are exaggeratingly illustrated in the inset). In **c**, the X-ray direction is perpendicular to the plane, and magnetic moments are also slightly tilted to the out-of-plane direction (the corresponding perpendicular applied field is ~1.6 T). **d–f** Projected components of DOSs for $T$ moment ($D^T$), $S$ ($D^S$), and $L$ ($D^L$) (left to right) were obtained from the first-principle calculations. In **e**, the value of $D^S$ is 50 times magnified. The magnitude of $D^T_x$ is much larger than those of $D^S_x$ and $D^L_x$.

magnetization due to spin canting: the magnetic moments of sublattice B and C in Fig. 4b is slightly tilted to the X-ray directions to reproduce the situation with in-plane external magnetic field of ~1 T ($M_{||} \approx 18$ m$\mu_B$/f.u.). In Fig. 4c, the direction of incident X-ray is perpendicular to the plane, and magnetic moments are slightly tilted towards out-of-plane direction

($M_\perp \approx 30$ m$\mu_B$/f.u., the corresponding perpendicular applied field is ~1.6 T). The XMCD spectra of each sublattice are also shown in the figure: the individual XMCD contributions from sublattice A, B, and C are illustrated by magenta, dashed light green, and blue lines. In the calculation for Fig. 4a, the Mn spin moments are perfectly compensated, so that the spin contribution for XMCD is

expected to be zero. However, a clear large XMCD signal appears. The appearance of the XMCD signal can be explained as follows. In Fig. 4a, the shape and amplitude of XMCD signals are the same for sublattice B and C (light green and blue lines), but are different between A and B (C) [magenta and light green (blue) lines]. Therefore, the observable total XMCD contribution remains, and the origin is attributed to the $T_z$ term[34]. The comparison between Fig. 4a, b shows that the in-plane XMCD spectral shape and intensity are mostly independent of the presence of small in-plane magnetization, except for the slightly reduced peak intensity at the $L_3$ edge and small changes of spectral shape in the $L_2$ edge. This feature also supports that the main contribution of in-plane XMCD is not arising from the spin magnetization but $T_z$ moment. In Fig. 4c, the XMCD response is smaller than that of Fig. 4a. Note that the sign of XMCD in Fig. 4c is opposite and spectral shape is different compared with the parallel configuration (Fig. 4a, b). These features confirmed by our spectral calculation are qualitatively consistent with the experimental observations, i.e., external-field-independent in-plane XMCD signal, and different spectral sign and shape between parallel and perpendicular directions.

The large discrepancy in XMCD intensity between the calculation and experiment (the intensity obtained by the model calculation is about 20% of XAS, but in experiment, the XMCD is only 0.2–0.4%) is explained as follows. In our model calculation, only the $x^2-y^2$ type orbital is considered, thus the orbital quenching is not properly expressed, i.e., the orbital angular moment is much overestimated. Since the $T_z$ term is roughly proportional to the product of spin and angular momenta[21], the intensity of $T_z$ mediated XMCD is much overestimated in our model calculation. In actual Mn₃Sn, the intensity of XMCD/XAS is much suppressed compared with the model calculation since the orbital angular moment is nearly quenched. There are also some differences in spectral shape between the model calculation and experiment. Especially, the double minimum structure observed in the experiment is not reproduced by our model calculation. This might be originating from the itinerant nature of this material, while the model calculation is based on the localized picture. Further improvement of spectral calculation is needed for a complete understanding of spectral shape in this material.

We have also measured in-plane field angle dependence of XMCD signal as shown in Supplementary Fig. S2b. As can be seen in the figure, the spectral shape is mostly independent of in-plane field direction. This behavior is also consistent with the model calculation for the case when the field and incident X-ray are parallel to the [01-10] axis, as shown in Supplementary Fig. S2d. This behavior is due to the inverse rotation of sublattice moments with external magnetic field rotation[18,35].

We will also show the result of model calculation when the spin chirality is positive, in Supplementary Fig. S2e. In this case, three XMCD contributions from each sublattice are completely compensated since the relative configurations between spin and orbital for A, B, and C sublattices are essentially same. These results obtained by our model calculation are qualitatively consistent with the prediction from the group theory[18]. Note that the magnetic structure with positive spin chirality (inset of Supplementary Fig. S2e) is not formed in real Mn₃Sn, and a fictitious magnetic structure for comparison. Since it is predicted that the ferroic octupole order only appears when the spin chirality is negative[17], the model calculation shows a complete correspondence between the presence of the XMCD signal and octupole order. In the XMCD experiment, although the direct detection of octuple may not be allowed, $T_z$-mediated detection is proposed by recent theoretical investigations[18,22].

To see the relationship between the microscopic feature of DOSs and the XMCD spectrum in detail, we have also performed a first-principle calculation. Figure 4d–f shows the obtained projected DOSs of anisotropic dipole moment $T$ ($\mathbf{T} = \mathbf{S} - 3\mathbf{r}(\mathbf{r} \cdot \mathbf{S})$) ($D^T$), $S(D^S)$, and $L(D^L)$ for each principal axis (see "Methods" section in detail). As can be seen in the figure, the amplitude of $D^T_x$ is much larger than that of $D^S_x$ and $D^L_x$. Moreover, the shape of $D^T_x$ near the Fermi energy is similar to that of $L_3$ edge of XMCD. This is consistent with the relation between the XMCD spectra and DOSs through the sum rule (also see "Methods" section), and shows that the present XMCD originates from the magnetic dipole moment in Mn. The agreement of XMCD spectra between theory and experiment for $L_2$ edges is rather poor, and might be coming from ignored transition probability in this relation (Supplementary Fig. S3).

In conclusion, we have both experimentally and theoretically demonstrated that the unconventional XMCD in Mn₃Sn is accompanied by the inverse triangle AF order characterized by the TRS-broken cluster magnetic octupole moment. The mechanism is the coupling between circularly polarized X-ray and the magnetic dipole moment called $T_z$ term, which is not compensated in the inverse triangle AF order, i.e., the spin chirality is negative. Our present study experimentally establishes the relationship between XMCD signal and magnetic octupole order. Although the XMCD detection of magnetic octupole moment was predicted theoretically, our experimental observation of magnetic octupole order shows the efficiency of XMCD to detect ferroic higher-rank multipole order. This result may expand the applicable target of X-ray spectroscopy. For example, the spatial order and correlation of the higher-rank multipoles, i.e., antiferroic higher-rank multipole order and the short-range order, can be probed by the resonant X-ray scattering, which becomes a technique combining the XMCD with X-ray diffraction. These observation techniques will accelerate the fundamental understanding of multipole physics. Especially TRS-broken higher-rank multipole orders will be revealed as a microscopic origin of nontrivial physical properties.

## Methods

**Single crystal growth and characterization**. The single crystals were grown by the solution-Bridgman method, using principles similar to ref. [36]. Polycrystalline precursors were synthesized by heating Mn (99.99%, Rare Metallics) and Sn (99.999%, Rare Metallics) at a ratio of 2.9:1 in an alumina crucible within an evacuated quartz ampoule at a temperature of 1050 °C for 6 h. The precursor ingots were transferred to a single-zone vertical Bridgman furnace with a central temperature of 1100 °C. The melted precursor was directionally solidified by passing it through the temperature gradient at a rate of 0.25 mm/h, and the resulting ingot contains two well-separated regions: single crystals of Mn₃Sn and a Sn-rich eutectic. Powder X-ray diffraction (Cu Kα, SmartLab, Rigaku) confirmed the phase purity of the isolated Mn₃Sn single crystals. The crystals were oriented using a Laue diffractometer and cut using spark erosion. Composition analysis using ICP-OES indicated that the sample used in the experiment has a composition of approximately Mn₃.₀₁₆Sn₀.₉₈₄.

**XMCD experiments**. XMCD experiments were performed at synchrotron beamlines BL25SU and BL-16A at SPring-8 and Photon Factory (KEK), respectively. The sample surface was obtained by cleaving (0001) plane of Mn₃Sn in a high vacuum. The angles between circular polarized X-ray beam and magnetic field from the (0001) surface are defined as $\theta_{in}$ and $\theta_B$, respectively. The X-ray absorption was detected by the total electron yield, and XMCD spectra were obtained from the difference between right and left circular polarizations and magnetic field reversal. The photon energies were swept from 630 to 670 eV or a higher energy.

**Single-atom XMCD-spectrum calculations**. XAS and XMCD spectra in the antiferromagnetic state of Mn$^{+1}$ within the $D_{2h}$ crystal electric field (CEF) were calculated with the following Hamiltonian[34]

$$H_i = H_{atom} + H_{CEF} + H_{MF}, \qquad (3)$$

where index $i$ denotes a site of the triangular lattice. The first term $H_{atom}$ is described as

$$
\begin{aligned}
H_{atom} = {} & \epsilon_d \sum_\gamma d_\gamma^\dagger d_\gamma + \zeta_{3d} \sum_{\gamma 1, \gamma 2} (\mathbf{l} \cdot \mathbf{s})_{\gamma 1, \gamma 2} d_{\gamma 1}^\dagger d_{\gamma 2} \\
& + \epsilon_p \sum_\gamma p_\gamma^\dagger p_\gamma + \zeta_{2p} \sum_{\gamma 1, \gamma 2} (\mathbf{l} \cdot \mathbf{s})_{\gamma 1, \gamma 2} p_{\gamma 1}^\dagger p_{\gamma 2} \\
& + \frac{1}{2} \sum_{\gamma 1, \gamma 2, \gamma 3, \gamma 4} g_{dd}(\gamma 1, \gamma 2, \gamma 3, \gamma 4) d_{\gamma 1}^\dagger d_{\gamma 2} d_{\gamma 3}^\dagger d_{\gamma 4} \\
& + \frac{1}{2} \sum_{\gamma 1, \gamma 2, \gamma 3, \gamma 4} g_{dp}(\gamma 1, \gamma 2, \gamma 3, \gamma 4) d_{\gamma 1}^\dagger d_{\gamma 2} p_{\gamma 3}^\dagger p_{\gamma 4},
\end{aligned}
\tag{4}
$$

where $d_\gamma^\dagger$ represents the creation operator for a $3d$ electron, including a combined index $\gamma$ with orbital and spin, and $p_\gamma^\dagger$ represents the one for a $2p$ core hole. $H_{atom}$ includes $3d$ level ($\epsilon_d$), spin–orbit coupling constant for $3d$ orbital ($\zeta_{3d}$), $2p$ level ($\epsilon_p$), spin–orbit coupling constant for $2p$ orbital ($\zeta_{2p}$), Coulomb interaction between $3d$ states ($g_{dd}$), and Coulomb interaction between $3d$ and $2p$ states ($g_{dp}$). These spin–orbit coupling constants and Slater integral for $g_{dd}$ and $g_{dp}$ were estimated from the ionic calculation within the Hartree–Fock–Slater method[37]. The Slater integral were reduced down to 80%, as in the previous studies[38,39], and the value of the spin–orbit coupling constant $\zeta_{3d}$ was zero because of describing a simple electron state under Lussell–Saunders coupling. The second term $H_{CEF}$ is satisfied with a one-electron potential of $D_{2h}$ symmetry, and the one-electron potential is expressed as

$$
\begin{aligned}
V_{crys} = {} & B_0^2 C_0^{(2)} + B_2^2 (C_2^{(2)} + C_{-2}^{(2)}) \\
& + B_0^4 C_0^{(4)} + B_2^4 (C_2^{(4)} + C_{-2}^{(4)}) + B_4^4 (C_4^{(4)} + C_{-4}^{(4)}).
\end{aligned}
\tag{5}
$$

From our first-principle calculation of Mn$_3$Sn, we estimated the CEF parameters to be $B^2_0 = 0.00$, $B_2^2 = 0.01$, $B^4_0 = 0.02$, $B_2^4 = 0.33$, and $B^4_4 = -0.32$ in a unit of eV, where $x^2 - y^2$ type orbital locates in lowest energy level. The third term $H_{MF}$ is expressed as

$$
H_{MF} = \sum_{\gamma 1, \gamma 2} \left( \mathbf{h}_{MF}^{(i)} \cdot \mathbf{s} \right)_{\gamma 1, \gamma 2} d_{\gamma 1}^\dagger d_{\gamma 2},
\tag{6}
$$

where $\mathbf{h}_{MF}^{(i)}$ denotes the molecular field for the spin part of the $i$ site of the triangular lattice.

**First-principles calculation.** First-principles calculations were performed with the QUANTUM ESPRESSO package[40]. We used the projector augmented-wave (PAW) pseudopotentials with the functional type of Perdew, Burke, and Ernzerhof[41] and with fully relativistic effects. The lattice constants $a = 5.665$ Å and $c = 4.531$ Å were adopted from the experiment[15].

The projected DOSs for $T$ moment, $S$ and $L$, are defined as

$$
D^A(E) = \sum_{i,\mathbf{k}} \delta(E - \epsilon_{\mathbf{k},i}) \cdot \langle \psi_{\mathbf{k},i} | \hat{A} | \psi_{\mathbf{k},i} \rangle,
\tag{7}
$$

where $A$ represents $T(\mathbf{T} = \mathbf{S} - 3\mathbf{r}(\mathbf{r} \cdot \mathbf{S}))$, $S$, or $L$, and $\epsilon_{\mathbf{k},i}$ and $\psi_{\mathbf{k},i}$ are the $i$th eigenvalue and eigenvector at the wavenumber $\mathbf{k}$, respectively. To calculate these projected DOSs, we constructed the tight-binding model by using Wannier functions with WANNIER90 package[42]. The tight-binding model was generated to reproduce the DFT energy band with Mn $s$, $p$, $d$, and Sn $s$, $p$ orbitals. The operators of $T$, $S$, and $L$ are defined for Wannier functions corresponding to the Mn $d$ orbitals, and the expected values of $T$, $S$, and $L$ were calculated with the eigenfunctions obtained from the tight-binding model.

The relation between the XMCD spectra and the projected DOSs was derived by the sum rules[42,43]. Namely, assuming that the sum rules hold at any band filling with the right band picture and that the XAS, $r$, be proportional to the number of holes, $n_h$[43,44], we can obtain the differential formulation of the sum rules as

$$
\frac{\sigma_{L_3}(E) + \sigma_{L_2}(E)}{r} = \frac{1}{2} \frac{D^L(E)}{n_h},
\tag{8}
$$

$$
\frac{\sigma_{L_3}(E) - 2\sigma_{L_2}(E)}{r} = \frac{\frac{2}{3} \cdot D^S(E) + \frac{7}{3} \cdot D^T(E)}{n_h}.
\tag{9}
$$

Here $\sigma_{L_3}$ and $\sigma_{L_2}$ are the XMCD spectra at the $L_3$ and $L_2$ edge, respectively, and $\Omega = r/n_h$ is a constant. Using Eqs. (8) and (9), we calculated the XMCD spectra in Supplementary Fig. S3.

## Data availability
The data that support the findings of this study are available from the corresponding author upon reasonable request.

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

## Acknowledgements

We are grateful to H. Kusunose, S. Hayami, M. Mizumaki, and M.-T. Suzuki for fruitful discussions. This work was partly supported by JSPS KAKENHI with project Nos. 19K03736, JP16H05990, JP19H04399, and by MEXT Quantum Leap Flagship Program (MEXT Q-LEAP) Grant Number JPMXS0120184122, and by Research Foundation for Opto-Science and Technology. This work is also partially supported by PRESTO (JPMJPR177A), CREST(JPMJCR18T3), Japan Science and Technology Agency (JST), and by Grants-in-Aids for Scientific Research on Innovative Areas (15H05882, 15H05883, and 15H05885) from the Ministry of Education, Culture, Sports, Science, and Technology of Japan, and by Grants-in-Aid for Scientific Research (16H06345, 18H03880, 19H00650), and by Center for Science and Innovation in Spintronics (CSIS) Tohoku University. Work at the Institute for Quantum Matter, an Energy Frontier Research Center was funded by DOE, Office of Science, Basic Energy Sciences under Award # DE-SC0019331. The experiments were performed at SPring-8 with proposal Nos. 2018A1525, 2018B1533, 2019A1589, and also carried out at Photon Factory (KEK) with proposal Nos. 2018S2-006, and 2018PF-31.

## Author contributions

M.K. planned the experimental project, and M.K., Y. Yamasaki, C.T., Y. Yokoyama, Y.K., K.A., H. Nojiri, H. Nakao, and T. N. performed XMCD experiments and data analysis. N.S., K.K., and T.K. performed theoretical calculations. M.I., T.T., and S.N. prepared single crystal samples. Y. Yamasaki, H. Nakao, and T.-H.A. proposed a basic concept to interpret the experimental data. M.K., N.S., K.K., T.K., H. Nakao, and T.-H.A. wrote the manuscript with comments from all co-authors.

## Competing interests

The authors declare no competing interests.
