## [Peer Review File · Nature Communications]

Editorial Note: Parts of this peer review file have been redacted as indicated to remove third-party material where no permission to publish could be obtained.

REVIEWER COMMENTS

Reviewer #1 (Remarks to the Author):

In the manuscript "X-ray study of ferroic octupole order producing anomalous Hall effect" by Kimata et al., the authors report on an anisotropic XMCD in the noncollinear antiferromagnet Mn₃Sn. Combined with theoretical analysis, they claim that the experimentally observed XMCD signals are originating from the cluster magnetic octupole order instead of the small spontaneous magnetization in Mn₃Sn. As the theoretical study in the possibility to detect the cluster magnetic octupole order by XMCD was already reported in Ref.[15], the new contribution of this work would be an experimental proof. Although the authors have established a connection between the XMCD signal and magnetic octupole order to some extent, I find some results of the study lack solidity and depth. Due to this, I do not recommend it for publication in Nature Communications.

1. In Fig. 1c, the peak position of XMCD located at slightly lower than that of XAS, but there is no further explanation for this phenomenon in the manuscript. It's may be important to distinguish the source of the signal.
2. Why there are two peaks in the L2 edge for $\theta_B = 15^\circ$ and the L3 edge for $\theta_B = 90^\circ$ and $B = 1$ T (Fig. 2a)?
3. The authors claim that the XMCD signal is proportional to the projection component parallel to the incident X-ray. Based on this analysis, there should be a small but not negligible XMCD signal for $\theta_{in} = 100^\circ$, $\theta_B = 90^\circ$ and $B = 0$ T because $\cos(100^\circ) \approx -0.17$ (Fig. 2a).
4. Related 3, if the XMCD signal for $\theta_{in} = 100^\circ$ and $\theta_B = 90^\circ$ contains some contribution from the magnetic moment in plane, the calculated moment is not accurate.
5. In the field swept data at a constant photon energy, there is a hysteresis for $\theta_B = 15^\circ$ (Fig. 2c). For the noncollinear antiferromagnet Mn₃Sn, one can switch the staggered moment direction of the triangular spin structure by rotating the net ferromagnetic moment by external magnetic field. Therefore, the coercivity of the hysteresis in Fig. 2c should be consistent with that of the magnetization loop in Extended data Fig. 1a, but it does not.
6. Why does the XMCD intensity at the L2, L3 edge of sample #3 is three times smaller than that of sample #2?
7. What's the direction and magnitude of external magnetic field in Fig. 4a and e?
8. It is stated that "the XMCD essentially independent of the in-plane field direction" in the antepenultimate paragraph. It makes me confused because the XMCD intensity is obviously related to the external field in Fig. 2c. Even the authors wrote "This behavior is due to the inverse rotation of sublattice moments with external magnetic field rotation" after the sentence.
9. In the figure caption of Fig. 4e, the authors attribute the intensity variation between different in-plane angle to the surface oxidation. But there is no evidence to support the conclusion. Besides, the intensity of XMCD signal increases with the in-plane angle. Additional experiments are needed to clarify the phenomenon.

Reviewer #2 (Remarks to the Author):

In this manuscript the authors report the experimental evidence of the magnetic octupole order in Mn₃Sn, probed using X-ray magnetic circular dichroism (XMCD). They find that the XMCD signals decouple to the static magnetization: while the effective spin moment m_{seff} obtained from the XMCD sum rules is consistent with the static magnetization for $B//[0001]$, m_{seff} for the fields close to the in-

plane direction disagrees with static magnetization, instead it is negative and independent of magnetic field strength. These observations are sharply contrasted with the conventional XMCD signals which arises from the spontaneous magnetization, and suggests the m_{eff} probed for fields oriented close to the in-plane direction does not originate from spin contribution. Further, they performed spectral calculations, from which they found the anomalous XMCD signal probed in Mn₃Sn could be attributed to the time-reversal symmetry broken magnetic octupole order. These results provide evidence for the theoretically predicted multipole order in Mn₃Sn. I think this work is interesting, novel and can be considered for publication in Nature Communications. It may deepen understanding of topological properties of this material, which attracted extensive attention.

However, I note the current version of the manuscript have some issues, which must be addressed before acceptance.

1. The authors mentioned that "the peak intensity of the XMCD signal is approximately 0.5~1% of the absorption. This value is small compared with that of the typical 3d ferromagnetic metals, but still too large to be explained by the small spontaneous magnetization of Mn₃Sn as discussed below". The authors should give the value of peak intensity of the XMCD signal of conventional FM and AFM metals for references. Given the authors have performed the spectral calculations for three magnetic sublattices and obtained the total XMCD spectrum by taking their sum, quantitative comparison between theory and experiment should be presented. Such a comparison could significantly strengthen the authors' major claim that the XMCD parallel to the Kagome plane originates from dipole term.

2. In figure 3(a), the m_{eff} and m_L obtained from the ferrioc sum rules are negative in the whole field range for the field orientation angle of 15 deg. The author stated "these results allow us to conclude that the XMCD parallel to the kagomé plane arises from the magnetic dipole term, which is unobservable in a static magnetization measurement". This discussion is not clear to readers who are not in this area. Why m_{eff} and m_L become negative in the presence of the magnetic dipole? Can the spectral calculations predict these results?

3. The authors argue the ferrioc order of magnetic octupole revealed by XMCD studies clearly demonstrates that the exotic material functionalities are closely related to the multipole order and even used the title "X-ray study of ferrioc octupole order producing anomalous Hall effect". I think this is overstated. In my opinion, this manuscript reports only experimental evidence of magnetic octupole order obtained from the XMCD experiments, but never discusses how the magnetic octupole order results in anomalous Hall effect.

Reviewer #3 (Remarks to the Author):

This manuscript describes an XMCD study on Mn₃X, a material with interesting magnetic properties and is predicted to have a cluster magnetic octupole moment. The major finding is based on the observation of an XMCD signal and its origin. The presented XMCD data show clear anisotropies, and two different origins as shown by the different spectral shapes. From XMCD sumrule applied to the data (including angular dependence arguments), expectation values of the orbital, spin and magnetic dipole operator are extracted.

One signal is interpreted in terms of the intraatomic magnetic dipole moment T_z component in the absence of a significant spin component, that is claimed to be directly related to the occurrence of a ferromagnetic order of magnetic cluster octupoles.

First principle atomic and bands structure calculations are compared to the observed spectral shapes

and are used to support this interpretation.

This is clearly an interesting interpretation of the data. It has been already predicted that the T_z term could be coupled to such a magnetic octupole moment in this system [Ref. 15]. Such a finding would be indeed very interesting for a broader community and its worth to be considered in the interpretation of the data.

There are however, several significant doubts on this interpretation that has to be overcome before I could support publication. In addition, the presentation and overall presentation of the results should be improved to make it easier to the reader understand the results.

A) A particular problem is that the temperatures are not stated at which the XMCD spectra are taken, except for figure 3. Does this mean the effect is only there at room temperature? Is it claimed the material has a ferro-cluster octupole moment above T_N , and is represented by the XMCD hysteresis in the magnetisation? Can that be more clearly stated?

If so, how does this go with the model picture of the moment directions and the calculation?

If the spins in average are disordered, how can this be model with three spin configurations? This must be properly explained.

B) Calculations and Fig. 4

I do not understand how the signal could depend on the chirality of the magnetic structure. This does not make any sense from symmetry point of view, in particular as the magnetic structure is incommensurate or not ordered.

In addition, the magnetic structure shown in figure 4 a and b, though they have opposite chirality, they have completely different energies and are not degenerate.

Note that only the second one has a head to tail structure the first one is energetically very unfavorable for next neighbours AFM exchange. Anyway opposite chirality would require to flip all spins, or one has to state that one compares two different magnetic structures. It must be explained why this would be the basic magnetic structure within the triangle. This is all unclear.

C) Also the results of the sum rules are very uncommon.

ML is clearly negative and beyond zero for $\theta_B = 15^\circ$. But zero for $\theta_B = 90^\circ$.

However, one would assume that m_l and s_z are coupled. I mean that S_z/L_z ratio is usually a constant. Also being directionally parallel or antiparallel.

Are you claiming that m_l is coupled to T_z instead? This would be rather strange, would it not? It needs to be discussed.

C) Can it be explained by defects or an impurity phase that is exchange biased when AFM sets in? can one exclude that? It would explain the disappearance, the weak signal and the different shape (maybe insulating) as well as the larger ML.

There are some minor and technical comments below:

1) Line 2 injections of spin is misleading, as photons have no spin and also injection is a bit strange, may be transfer would be better. I assume you want to say that there is a magnetic angular momentum transfer from the photon to the sample allowing to probe magnetism.

2) Line 9 Its not the magnetic moments that are 0 but the magnetization

3) Figure 2: It is very difficult to understand all panels of Figure 2 from the info provided in the figures and its caption. It might be helpful to always specify both, θ_{in} and θ_B in all panels. In 2c) After looking at it for some time it, the field direction in the caption might be the field sweep direction indicated by the arrows? Can one put all the info in the caption and the figure so it is self-explaining?

4) It is written a large ferroic response is observed (fig. 2 caption), but I do not see any ferroic response, there is no opening of the hysteresis which is required for a ferroic response. It's not compatible with the magnetization data.

5) Why are the sample #1, #2, #3, are they different in any way? Is the signal time dependent or any reason why one uses different samples? One should comment on that.

6) Text to Fig. 2 (remanent component) please describe field history of the zero field measurement then. If there is no field history, don't call it remanent. It might also be helpful to add also in the inset the 0 T data.

7) There is a sentence ..."see the sum rule analysis shown in bellow"
may should be "shown below"?

8) It might be helpful to label the two different XMCD responses (e.g. different B vs crystal axis and spectral shapes differently)

9) Why is the theta 15 degree not a sum of an XMCD of $\langle s \rangle$ and $\langle T_z \rangle$? , why not visible at theta 15? (How does the difference plot between 1T and 0T, looks like for theta 15 degrees?), as the magnetization is similar for both directions at 1 T.

10) Also Sentence: "This highly anisotropic behavior of XMCD is more clearly confirmed in the field swept data at a constant photon energy" anisotropic in what? Connected to the previous point.
Improve formulation

11) I don't understand the connection between:
"XMCD mostly vanishes at $T = 200$ K." and "the XMCD signal detect the bulk properties."

12) There is an inconsistency between the calculation and the data. Though the overall shape seems in reasonable agreement with the data, the maxima in the calculation in XMCD is coming on the maximum of the absorption spectra, on not on the inflection point. This means that these are clearly are not the same states or they have different energies. In other words the XMCD L3 spectra looks more like a derivative of the L3 spectra than whereas the calculation does not. Of course, these calculations are not very precise, but then they are not such a strong evidence that of the origin of the signal.

Dear Reviewers,

Thank you very much for the reviewers' valuable comments, which are very helpful to improve our manuscript. Here, let us reply to reviewers' comments. We believe that the revised manuscript is now worthwhile publishing in Nature Communications.

Best wishes,

Authors

REVIEWER COMMENTS

=====

Reviewer #1 (Remarks to the Author):

In the manuscript "X-ray study of ferroic octupole order producing anomalous Hall effect" by Kimata et al., the authors report on an anisotropic XMCD in the noncollinear antiferromagnet Mn₃Sn. Combined with theoretical analysis, they claim that the experimentally observed XMCD signals are originating from the cluster magnetic octupole order instead of the small spontaneous magnetization in Mn₃Sn. As the theoretical study in the possibility to detect the cluster magnetic octupole order by XMCD was already reported in Ref.[15], the new contribution of this work would be an experimental proof. Although the authors have established a connection between the XMCD signal and magnetic octupole order to some extent, I find some results of the study lack solidity and depth. Due to this, I do not recommend it for publication in Nature Communications.

Reply:

Thank you for reviewer's understanding to our experimental proof of x-ray detection of magnetic octupole. To reinforce our scenario, we have newly added some spectral calculations as shown in Fig.4 in the revised manuscript. As a result, we have succeeded in reproducing main features of experimental observations, i.e., nearly field-independent intensity and shape of in-plane XMCD signal, and different spectral sign and shape between parallel and perpendicular field directions. These features cannot be explained by conventional magnetization origin of XMCD, but can be explained by T_z mediated XMCD,

which is closely related to the magnetic octupole order. Thus, we are considering that our manuscript is now more solidly.

Below, we would like to make point by point reply to the reviewer's comments.

* There were some lacks in reference numbers, and wrong labeling between sample number and experimental data in the original manuscript, so we have corrected ref numbers and sample numbers in the revised version. We would like to apologize for these confusing.

1. In Fig. 1c, the peak position of XMCD located at slightly lower than that of XAS, but there is no further explanation for this phenomenon in the manuscript. It's may be important to distinguish the source of the signal.

Reply:

From our experimental results, we are considering that the low-energy peak of L_3 edge in XMCD is one of the features in the present signal. According to our recent theoretical investigation [ref32 in the revised manuscript, N. Sasabe, M. Kimata, and T. Nakamura, Phys. Rev. Lett. **126**, 157402 (2021)], the peak position of T_z mediated XMCD could depend on the orbital configuration. In our present paper, although the x^2-y^2 type orbit is set to the lowest energy and localized picture is assumed, this assumption might be too simplified for the complete explanation of the experimental situations. Indeed, the first principles calculation suggests that the lowest-energy orbital for minority spin is a mixture of x^2-y^2 and z^2-r^2 with a ratio of $\sim 8:2$ due to the metallic nature of this material. Thus, we are considering that the situation of our present model calculation is reasonably correct as the first approximation, but further improvement is needed for better understanding of spectral shape.

In the revised manuscript, we do not emphasize this point and deleted corresponding descriptions. Instead of this point, we have added descriptions to stress the difference of spectral shape between parallel and perpendicular field (and also incident x-ray) directions.

➤ **Deleted description in the original manuscript:**

" Moreover, the XMCD peak positions located at slightly lower energies of XAS peaks are another characteristic of the present XMCD signal."

➤ **Revised descriptions (lines 119-122 in the revised manuscript):**

" Note that the XMCD sign and spectral shape is totally different between $\theta_B = 15^\circ$ and 90° . For example, the strongest peak position at the L_3 edge for $\theta_B = 15^\circ$ (red arrow) is slightly lower than that for $\theta_B = 90^\circ$ (blue arrow), and the shape of L_2 edge has double minimum for $\theta_B = 15^\circ$, while that for $\theta_B = 90^\circ$ shows a broad peak."

2. Why there are two peaks in the L2 edge for $\theta_B = 15^\circ$ and the L3 edge for $\theta_B = 90^\circ$ and $B = 1$ T (Fig. 2a)?

Reply:

For the first question, the XMCD for $\theta_B = 15^\circ$ is arising from the T_z term, but it is not easy to simply explain the origin of double peaks in L_2 edge at the moment since the spectral shape purely coming from the T_z term may not be well understood.

Based on our model calculations, the spectral shape of present XMCD is originating from the sum of three sublattice contributions. But the agreement of spectral line shape between theory and experiment is not so good. For instance, the double peak structure of L_2 edges is not reproduced. This discrepancy might be coming from the itinerant feature of this material, while the spectral model calculation is based on a localized single ion picture. Please note that the double-peak feature of L_2 edge is well reproduced for all measured samples. So, we are considering the observed XMCD feature in this material is intrinsic, but further studies including theoretical calculation is needed for the complete understanding of spectral shape. Related to this point, we have added following descriptions.

➤ **Lines 260-265 in the revised manuscript:**

" There are also some differences in spectral shape between the model calculation and experiment. Especially, the double minimum structure observed in the experiment is not reproduced by our model calculation. This might be originating from itinerant nature of this material, while the model calculation is based on the localized picture. Further improvement of spectral calculation is needed for complete understanding of spectral shape in this material."
"

Reply:

For the second question, the lower-energy peak at the L_3 edge for $\theta_B = 90^\circ$ is coming from the reduced in-plane XMCD signal since the incident x-ray has a small in-plane component correspond to $\cos 100^\circ \approx -0.17$, as pointed out by the reviewer's next comment. Indeed, as shown in the figure below, the lower-energy peak at the L_3 edged (dashed arrow in the left panel presented below) is much suppressed for $\theta_{in} = 90^\circ$ ($\theta_B = 80^\circ$), where the incident x-ray is perpendicular to the kagome plane (please see the blue curve of left panel shown below).

For $\theta_{in} = 90^\circ$, the overall intensity of XMCD is reduced due to the oxidation of sample surface during the experiment, which can be seen in the development of multiplet structures in XAS (arrows in the center panel). These multiplet structures are consistent with that of Mn^{2+} , suggesting the formation of MnO (right panel of the following figure from [B. Gilbert, *et. al.*, *J. Phys. Chem. A* **107**, 2839 (2003)]). Consequently, the obtained magnetic moment for $\theta_{in} = 90^\circ$ is suppressed to approximately half of bulk magnetization, so we would not add XMCD spectra for $\theta_{in} = 90^\circ$ in the manuscript.

From [B. Gilbert, *et. al.*, *J. Phys. Chem. A* **107**, 2839 (2003)]

[Redacted]

Related to this point, we have added following descriptions.

➤ **Lines 132-135 in the revised manuscript:**

"The lower energy minimum at the L_3 edge for $\theta_B = 90^\circ$ (blue dashed arrow) is originating from the reduced in-plane XMCD due to the small in-plane component of incident x-ray with $\theta_{in} = 100^\circ$ since $\cos 100^\circ \approx -0.17$."

3. The authors claim that the XMCD signal is proportional to the projection component parallel to the incident X-ray. Based on this analysis, there should be a small but not negligible XMCD signal for $\theta_{in} = 100^\circ$, $\theta_B = 90^\circ$ and $B = 0$ T because $\cos(100^\circ) \approx -0.17$ (Fig. 2a).

Reply:

Thank you for the comment. In the case for $\theta_{in} = 100^\circ$, the field direction is perpendicular to the kagome plane ($\theta_B = 90^\circ$). Thus, in principle, the perpendicular field does not induce the in-plane XMCD component if the field is perfectly aligned. Therefore, the in-plane XMCD signal is zero for both $\theta_{in} = 90^\circ$ ($\theta_B = 80^\circ$), and $\theta_{in} = 100^\circ$ ($\theta_B = 90^\circ$). However, in actual experiment, there might be slight misalignments of field direction, and this could induce in-plane signal if the field strength is sufficiently large, i.e., the field component parallel to the plane is larger than that to switch the in-plane magnetic moment. If we assume a misalignment of 1° , the in-plane field component induced by the external field of 1 T is expected to be $1 \text{ T} \times \sin(1^\circ) \approx 0.017 \text{ T}$. This value is larger than the in-plane switching field of present sample ($\sim 0.01 \text{ T}$). The effect discussed above is indeed observed in the spectrum for $B = 1 \text{ T}$ as shown in Fig. 2a (blue dashed arrow) in the revised manuscript.

Based on the reviewer's comment, we have plotted data points for $B = 1 \text{ T}$ in addition to 0.1 T. In the revised figure, the in-plane signal is very small for both $\theta_{in} = 90^\circ$ and 100° at 0.1 T, while the small negative in-plane signal is observed for $\theta_{in} = 100$ at 1T as the reviewer pointed out.

➤ **Related part in the revised manuscript: Fig.2b.**

4. Related 3, if the XMCD signal for $\theta_{in} = 100^\circ$ and $\theta_B = 90^\circ$ contains some contribution from the magnetic moment in plane, the calculated moment is not accurate.

Reply:

We would appreciate reviewer's comment of this point. Since the XMCD spectrum for $\theta_{in} = 100^\circ$ contains some in-plane components, the calculated moment is not accurate as the reviewer pointed out, i.e., the calculated moments overestimate the magnetic moment. As the in-plane

magnetic moment for S_{eff} is around $-60 \text{ m}\mu_{\text{B}}/\text{f.u.}$, The amount of overestimation is estimated as $-60 \text{ m}\mu_{\text{B}} \times \cos(100^\circ) \approx -60 \text{ m}\mu_{\text{B}} \times -0.17 \approx 10 \text{ m}\mu_{\text{B}}/\text{f.u.}$ If we apply this value for $B = 1 \text{ T}$, the corrected out-of-plane magnetic moment is $\sim 20 \text{ m}\mu_{\text{B}}/\text{f.u.}$, which is good agreement with the magnetization measurement.

➤ **Related parts in the revised manuscript (lines 173-179 in the revised manuscript):**

" As mentioned above, the XMCD spectrum for $\theta_B = 90^\circ$ ($\theta_{\text{in}} = 100^\circ$) contains reduced in-plane signal since the incident x-ray has a small in-plane component correspond to $\cos 100^\circ \approx -0.17$. This contributes to the overestimation of magnetic moments for this angle. Since $m_{S_{\text{eff}}}$ for in-plane direction is $\sim -60 \text{ m}\mu_{\text{B}}/\text{f.u.}$ on average, the overestimate for $\theta_B = 90^\circ$ direction is estimated to be $\sim 10 \text{ m}\mu_{\text{B}}/\text{f.u.}$ Thus, the out-of-plane magnetic moment at 1 T is estimated to be $\sim 20 \text{ m}\mu_{\text{B}}/\text{f.u.}$ This value is reasonably consistent with the magnetization observed in static measurement [1] (see also extended data Fig.1a). "

5. In the field swept data at a constant photon energy, there is a hysteresis for $\theta_B = 15^\circ$ (Fig. 2c). For the noncollinear antiferromagnet Mn_3Sn , one can switch the staggered moment direction of the triangular spin structure by rotating the net ferromagnetic moment by external magnetic field. Therefore, the coercivity of the hysteresis in Fig. 2c should be consistent with that of the magnetization loop in Extended data Fig. 1a, but it does not.

Reply:

Thank you for the comment. The switching fields of this material are slightly sample dependent. For instance, the switching field reported in refs [1, 3, 6] [Nakatsuji, S., Kiyohara, N., & Higo, T., *Nature* **527**, 212–215 (2015), Ikhlas, M., *et al.*, *Nature Phys* **13**, 1085–1090 (2017), and Higo, T., *et al.*, *Nature Photon* **12**, 73–78 (2018)] differs between ~ 0.03 and $\sim 0.01 \text{ T}$, which might be originating from some extrinsic origin, e.g., domain wall pinning by impurities and/or defects. On the other hand, in our present experiment, the switching fields of XMCD and magnetization measurements with distinct samples are $\sim 0.01 \text{ T}$ and $\sim 0.04 \text{ T}$, respectively. Thus, the difference of switching field is reasonable variation due to the sample dependence.

➤ **Related parts in the revised manuscript (lines 145-149 in the revised manuscript):**

" Although the switching field observed in XMCD ($\sim 0.01 \text{ T}$) is smaller than that of static

magnetization measurement (~ 0.04 T, see extended data Fig.1a) with distinct sample. The discrepancy between XMCD and magnetization might be originating from sample dependence with extrinsic origin. The variation of switching field is indeed observed in previous reports [1, 3, 6]. "

6. Why does the XMCD intensity at the L₂, L₃ edge of sample #3 is three times smaller than that of sample #2?

Reply:

Thank you for the comment. In the original manuscript, there is a problem of data presentation related the signal normalization. In the revised manuscript, we have normalized both XAS and XMCD signals by the XAS peak intensity at the L₃ edge. After this treatment, the intensity variation between sample #1 and sample #3 is reduced to be ~ 1.5 - 1.7 times. Also, there were wrong labeling between sample number and data, so we have corrected sample labels in the revised manuscript. We would like to apologize for this confusing.

The experiments for these samples were performed by using distinct samples with different batch and in different facilities. Thus, there might be some difference of sample and experimental conditions although we are also considering that the intensities should be identical as suggested by the reviewer.

As shown in the extended data Fig. 2b and c in the revised manuscript, the normalized XMCD intensity is reduced in factor of ~ 2 due to the surface oxidation. This suggests that the XMCD intensity could be slightly varied by some other surface contaminations. In the experiment for sample #3, we have cooled the sample to 200 K first, and then the temperature is increased to 300 K. Thus, the surface contamination by cryogen absorption might be increased compared with other samples. Also, we would note that the intensity variation observed here (~ 0.1 - 0.2% of XAS intensity) is not so large for usual experiments for ferromagnets.

However, these discussions are speculation, thus at the moment, we do not have clear answer to explain the intensity variation between samples. We have added descriptions listed below.

➤ **Caption of Fig.3b, lines 591-594:**

" The XMCD intensity of Fig.3b (sample #3) is almost 1.5-1.7 times smaller than that of Fig.2a (sample #1). Although the origin of intensity variation is not clear, might be arising from extrinsic difference of sample conditions including surface contaminations and/or distinct sample batch. "

7. What's the direction and magnitude of external magnetic field in Fig. 4a and e?

Reply:

The direction of external field is parallel to the incident x-ray and field strength is 0.1 T. We have revised corresponding figure based on the reviewer's comments.

Also, we have moved in-plane field angle dependence of XMCD to the extended data Fig.2 in the revised manuscript.

➤ **Related part in the revised manuscript: Extended data Fig.2a and b (shown below).**

Also, we have added following sentence in the caption of extended data Fig.2 in the revised manuscript.

➤ **lines 649-650:**

" The directions of external magnetic field and x-ray are parallel to each other within the kagome plane. "

8. It is stated that "the XMCD essentially independent of the in-plane field direction" in the antepenultimate paragraph. It makes me confused because the XMCD intensity is obviously

related to the external field in Fig. 2c. Even the authors wrote “This behavior is due to the inverse rotation of sublattice moments with external magnetic field rotation” after the sentence.

Reply:

Thank you for the comment and we are sorry for the reviewer's confusing. The main point here is essentially the same spectral shape of XMCD. As shown in next reply, the intensity variation observed here is coming from the surface oxidation during the experiment, which can be seen in the development of multiplet structures in XAS.

9. In the figure caption of Fig. 4e, the authors attribute the intensity variation between different in-plane angle to the surface oxidation. But there is no evidence to support the conclusion. Besides, the intensity of XMCD signal increases with the in-plane angle. Additional experiments are needed to clarify the phenomenon.

Reply:

Thank you for the comment. We have added additional data of XAS in the extended data Fig.2c in the revised manuscript. In this sample, the spectrum were measured in the order of $\varphi = 68^\circ \rightarrow 90^\circ \rightarrow 42^\circ \rightarrow 0^\circ$. As can be seen in the right figure presented below, XAS has a smooth shape in fresh surface ($\varphi = 68^\circ$, and 90°), but the multiplet structures (arrows in the figure) gradually increase for $\varphi = 42^\circ$, and 0° . By comparing the reference, the shape is similar to the XAS of Mn^{2+} , suggesting the formation of MnO . This fact implies that the surface oxidation is induced during the experiment.

➤ **Related parts in the revised manuscript (Extended data Fig.2b and c):**

Also related to this point, we have added following descriptions in the revised manuscript.

➤ **Lines 653-659 (Caption of Extended data Fig.2.):**

" In this experiment, the data for $\varphi = 68^\circ$ was firstly measured, and then φ is changed to 90° , 42° , and 0° . Although a high vacuum of chamber pressure is kept, the slight oxidation of sample surface is hard to avoid due to the intensive x-ray irradiation, which is indeed observed as the growth of multiplet structures correspond to Mn^{2+} in XAS (arrows in extended Fig.2c), i.e., formation of MnO. Since this Mn oxide layer is expected to show no XMCD, the surface oxidation might cause intensity reduction of XMCD signals. "

=====

Reviewer #2 (Remarks to the Author):

In this manuscript the authors report the experimental evidence of the magnetic octupole order in Mn_3Sn , probed using X-ray magnetic circular dichroism (XMCD). They find that the XMCD signals decouple to the static magnetization: while the effective spin moment m_{seff} obtained from the XMCD sum rules is consistent with the static magnetization for $B//[0001]$, m_{seff} for the fields close to the in-plane direction disagrees with static magnetization, instead it is negative and independent of magnetic field strength. These observations are sharply contrasted with the conventional XMCD signals which arises from the spontaneous magnetization, and suggests the m_{seff} probed for fields oriented close to the in-plane

direction does not originate from spin contribution. Further, they performed spectral calculations, from which they found the anomalous XMCD signal probed in Mn₃Sn could be attributed to the time-reversal symmetry broken magnetic octupole order. These results provide evidence for the theoretically predicted multipole order in Mn₃Sn. I think this work is interesting, novel and can be considered for publication in Nature Communications. It may deepen understanding of topological properties of this material, which attracted extensive attention.

However, I note the current version of the manuscript have some issues, which must be addressed before acceptance.

Reply:

Thank you for the positive comment for our manuscript. Here, we would like to reply reviewer's scientific comments.

* There were some lacks in reference numbers, and wrong labeling between sample number and experimental data in the original manuscript, so we have corrected ref numbers and sample numbers in the revised version. We would like to apologize for these confusing.

1. The authors mentioned that "the peak intensity of the XMCD signal is approximately 0.5~1% of the absorption. This value is small compared with that of the typical 3d ferromagnetic metals, but still too large to be explained by the small spontaneous magnetization of Mn₃Sn as discussed below". The authors should give the value of peak intensity of the XMCD signal of conventional FM and AFM metals for references.

Reply:

Thank you for the comments. We have added references of conventional ferro and antiferromagnetic materials as refs 23 and 24, in the revised manuscript. For example, a Mn based ferromagnet Co₂MnGe shows large XMCD correspond to ~10-20% of XAS, whereas the XMCD of typical antiferromagnet MnPt is almost absent.

Also, in the original manuscript, there is a problem related the signal normalization. In the revised manuscript, we have normalized both XAS and XMCD signals by the XAS peak intensity at the L_3 edge. After this treatment, the XMCD intensity of this materials is 0.2-0.4% of XAS. This point is also corrected in the revised manuscript.

Related to this point, we have added following descriptions in the revised manuscript.

➤ **Lines 104-108 in the revised manuscript:**

" This value is small compared with that of typical 3d ferromagnetic metals (10-20% of XAS) [22, 23]. However, as discussed below, the XMCD signal in Mn_3Sn is still too large to be explained by the small spontaneous magnetization of Mn_3Sn , whereas the XMCD response of typical colinear antiferromagnet is almost absent [24]. "

Given the authors have performed the spectral calculations for three magnetic sublattices and obtained the total XMCD spectrum by taking their sum, quantitative comparison between theory and experiment should be presented. Such a comparison could significantly strengthen the authors' major claim that the XMCD parallel to the Kagome plane originates from dipole term.

Reply:

At the moment, we are considering that the quantitative intensity comparison between model calculation and experiment is difficult. Indeed, there is a large discrepancy of XMCD intensity between the model calculation and experiment: the XMCD intensity obtained by the model calculation is about 20% of XAS, but in experiment, the XMCD is only 0.2-0.4%. This large discrepancy in intensity can be explained by the orbital quenching. In our model calculation, only the x^2-y^2 type orbital is considered, thus the orbital quenching is not properly expressed, i.e., the orbital angular moment is much overestimated. Since the T_z term is roughly proportional to the product of spin and orbital angular momenta [eq.8 in ref. 18 in the manuscript, Oguchi, T., & Shishidou, T. Phys. Rev. B **70**, 024412 (2004)], the intensity of T_z mediated XMCD is much overestimated in our model calculation. On the other hand, in actual Mn_3Sn , since the orbital angular moment is nearly quenched, which is confirmed by both

XMCD sum rule analysis and first principles calculation, the intensity of XMCD/XAS is much suppressed compared with the model calculation.

Nevertheless, our model calculation could be compared with experiment in spectral sign and overall shape. We are considering that the features of present XMCD spectrum observed in experiment (opposite sign, field-independent, different shape between parallel and perpendicular x-ray incident directions) are reproduced by the model calculation.

Related to this point, we have added following descriptions.

➤ **lines 253-260 in the revised manuscript:**

" The large discrepancy in XMCD intensity between the calculation and experiment (the intensity obtained by the model calculation is about 20% of XAS, but in experiment, the XMCD is only 0.2-0.4%) is explained as follows. In our model calculation, only the x^2-y^2 type orbital is considered, thus the orbital quenching is not properly expressed, i.e., the orbital angular moment is much overestimated. Since the T_z term is roughly proportional to the product of spin and angular momenta [18], the intensity of T_z mediated XMCD is much overestimated in our model calculation. In actual Mn_3Sn , the intensity of XMCD/XAS is much suppressed compared with the model calculation since the orbital angular moment is nearly quenched. "

2. In figure 3(a), the m_s^{eff} and m_L obtained from the ferromagnetic sum rules are negative in the whole field range for the field orientation angle of 15 deg. The author stated "these results allow us to conclude that the XMCD parallel to the kagomé plane arises from the magnetic dipole term, which is unobservable in a static magnetization measurement". This discussion is not clear to readers who are not in this area. Why m_s^{eff} and m_L become negative in the presence of the magnetic dipole? Can the spectral calculations predict these results?

Reply:

In general, the spin component of XMCD is always positive, while the dipole term (T_z) can be both negative and positive. Since the m_s^{eff} is the sum of spin moment m_s and T_z moment, the negative m_s^{eff} observed in the experiment is attributed to the contribution of T_z term.

Based on our model calculation, the x^2-y^2 type orbital is chosen as the lowest energy orbital, which is predicted by the first principles calculation, and this situation explains the negative sign of T_z moment. On the other hand, if the z^2-r^2 type orbital is chosen for the lowest energy as a fictitious situation, the sign of T_z moment is calculated to be positive.

Thus, the observation of negative T_z moment is consistent with the spectral model calculation based on the orbital configuration predicted by the first principles calculation.

Related to this point, we have added some sentences in the revised manuscript as follows.

➤ **Lines 192-200 in the revised manuscript:**

" Thus, the origin of negative and field-independent $m_{S_{\text{eff}}}$ for $\theta_B = 15^\circ$ could not be ascribed to the spin contribution $\langle S_z \rangle$ since the sign of $\langle S_z \rangle$ should be always positive. On the other hand, the sign of dipole term $\langle T_z \rangle$ can be both positive and negative depending on the spin and orbital configurations. Thus, the observation of negative $m_{S_{\text{eff}}}$ allows us to conclude that the XMCD parallel to the kagomé plane arises from the magnetic dipole term, which is unobservable in a static magnetization measurement. Indeed, the negative dipole term, which correspond to the positive spectral shape at the L_3 edge, is reproduced by our spectral model calculations together with the orbital configuration determined by the first principles calculations as shown below. "

Reply:

For the m_L moment, we have improved sum rule analysis procedure in the revised manuscript, and m_L is mostly zero for whole field range in both field directions (see the revised figure presented below). This feature is consistent with the prediction from our first principles calculation where the orbital moment is mostly quenched.

Here, we would show the comparison between original and improved procedure of sum rule analysis. Since the XMCD signal of this materials is rather small ($\sim 0.4-0.2\%$ of XAS intensity), some artifact signals could be magnified compared with conventional ferromagnets where the XMCD signal intensity is much larger ($\sim 10-20\%$ of XAS signal). In our experiment, the difference of XAS for left and right circularly polarization ($\mu^- - \mu^+$) is taken for both positive and negative magnetic fields. And then, the difference of $\mu^- - \mu^+$ between positive and negative magnetic fields is calculated as a XMCD spectrum.

Below, we will show a typical example of analysis procedure. The upper shows the original, and lower shows the improved procedure. In the original procedure, $\mu^- - \mu^+$ data (left panel in the original procedure shown below) show step-like background between low energy and high energy regions. However, this step-like background is an artificial due to the difference of XAS intensity between left and right circularly polarized x-ray by some reasons since the average of $\mu^- - \mu^+$ should be zero in the high energy limit, and should show inversed shape with field reversal. Although this offset is mostly compensated by the field reversal, a slight offset at the high energy region will still remain in the final XMCD spectrum, and this causes additional contribution to the integrated intensity of spectrum. This additional contribution in high energy region is directly affect the value of m_L moment.

On the other hand, in the improved procedure, the XAS spectrum is normalized to have a same value in the high energy region (typically $E > 665$ eV). The resultant $\mu^- - \mu^+$ is shown in the lower panel. In this case the high energy value is mostly zero except for the oscillation behavior, which might be originating from the XAFS oscillations. Also, the shape of $\mu^- - \mu^+$ is mostly inverted by the field reversal. These features in $\mu^- - \mu^+$ are more likely than that obtained from the original procedure. By using the improved XMCD spectrum, the q value of integrated XMCD ($\propto m_L$) is much suppressed.

Original analysis procedure

Improved analysis procedure

We have replaced Fig3a by new figure obtained by improved analysis (please see below). Relatively large errors for $\theta_b = 15^\circ$ is coming from the uncertainty of q value due to the oscillation behavior in the high energy region. In the revised Fig3a, m_L moments for both $\theta_b = 15^\circ$ and 90° are mostly zero for whole field range. We would also note that the value of m_S^{eff} is almost independent of analysis procedure, meaning that negative sign and nearly field-independent behavior of m_S^{eff} is intrinsic. Although the sample number in the revised figure is changed, this is due to our wrong labeling between sample number and data in the original manuscript. We have also corrected this point.

3. The authors argue the froic order of magnetic octupole revealed by XMCD studies clearly demonstrates that the exotic material functionalities are closely related to the multipole order and even used the title "X-ray study of ferroic octupole order producing anomalous Hall effect". I think this is overstated. In my opinion, this manuscript reports only experimental evidence of magnetic octupole order obtained from the XMCD experiments, but never discusses how the magnetic octupole order results in anomalous Hall effect.

Reply:

Thank you for the comment. In this material, the anomalous Hall effect mediated by the octupole order is established theoretically [Ref 8 in the revised manuscript, Suzuki, M.-T., et al., Phys. Rev. B 95, 094406 (2017)]. This octupole order is closely related to the triangle magnetic structure with negative spin chirality. The point of this manuscript is that the relationship between octupole order and XMCD experiment have been clarified. Thus, we are also considering that the title of our manuscript is suitable to summarize our work.

On the other hand, we would agree with the reviewer's first suggestion. So, we have changed corresponding descriptions as follows.

➤ **Lines 301-302 in the revised manuscript:**

" Our present study experimentally establishes the relationship between XMCD signal and magnetic octupole order. "

=====

Reviewer #3 (Remarks to the Author):

This manuscript describes an XMCD study on Mn₃X, a material with interesting magnetic properties and is predicted to have a cluster magnetic octupole moment. The major finding is based on the observation of an XMCD signal and its origin. The presented XMCD data show clear anisotropies, and two different origins as shown by the different spectral shapes. From XMCD sumrule applied to the data (including angular dependence arguments), expectation values of the orbital, spin and magnetic dipole operator are extracted.

One signal is interpreted in terms of the intraatomic magnetic dipole moment T_z component in the absence of a significant spin component, that is claimed to be directly related to the occurrence of a ferromagnetic order of magnetic cluster octupoles.

First principle atomic and bands structure calculations are compared to the observed spectral shapes and are used to support this interpretation.

This is clearly an interesting interpretation of the data. It has been already predicted that the T_z term could be coupled to such a magnetic octupole moment in this system [Ref. 15]. Such a finding would be indeed very interesting for a broader community and its worth to be considered in the interpretation of the data.

There are however, several significant doubts on this interpretation that has to be overcome before I could support publication. In addition, the presentation and overall presentation of the results should be improved to make it easier to the reader understand the results.

Reply:

Thank very much for the positive attitude of reviewer. Based on the reviewer's comments, we have improved data presentation to clearly show the experimental and theoretical results.

Below, we would like to reply to reviewer's scientific comments.

* There were some lacks in reference numbers, and wrong labeling between sample number and experimental data in the original manuscript, so we have corrected ref numbers and sample numbers in the revised version. We would like to apologize for these confusing.

A) A particular problem is that the temperatures are not stated at which the XMCD spectra are

taken, except for figure 3. Does this mean the effect is only there at room temperature? Is it claimed the material has a ferro-cluster octupole moment above T_N , and is represented by the XMCD hysteresis in the magnetisation? Can that be more clearly stated?

If so, how does this go with the model picture of the moment directions and the calculation? If the spins in average are disordered, how can this be model with three spin configurations? This must be properly explained.

Reply:

Thank you for the comments. As suggested by the reviewer, we have added measured temperatures for all figures. Basically, the reviewer's understanding is correct. This material shows triangle antiferromagnetic with negative spin chirality between 430 K and 270 K. The anomalous Hall effect (and also the ferro-cluster octupole order) only appears in this temperature range, i.e., $270 < T < 430$ K. Above, 430 K, this material is a paramagnet, and below 270 K, another magnetic phase appears although the detail of low-temperature phase is not well investigated.

Related to this point, we have added descriptions to explain the magnetic phases of this material.

➤ **Lines 205-210 in the revised manuscript:**

" In the well-ordered Mn_3Sn single crystals, the triangle magnetic structure with negative spin chirality is formed below 430 K, and another magnetic transition around 270 K was reported [29]. In the low-temperature phase below 270 K, the AHE and small spontaneous magnetization vanishes; it indicates that the TRS is preserved. [29] (see also extended data Fig. 1b). This means that the ferroic octupole order is only predicted between 270 K and 430 K. "

Reply:

As explained above, the magnetic structure between 270 and 430 K is well established [For instance, see ref [1] and [15] in the revised manuscript. [1] Nakatsuji, S., Kiyohara, N., & Higo, T. *Nature* **527**, 212–215 (2015), and [15] Tomiyoshi, S., *J. Phys. Soc. Jpn.* **51**, 803-810 (1982)]. Thus, in the model calculation of XMCD spectrum, we have chosen a unit of this magnetic structure which is characterized by triangle antiferromagnetic order with negative spin chirality.

B) Calculations and Fig. 4

I do not understand how the signal could depend on the chirality of the magnetic structure. This does not make any sense from symmetry point of view, in particular as the magnetic structure is incommensurate or not ordered.

In addition, the magnetic structure shown in figure 4 a and b, though they have opposite chirality, they have completely different energies and are not degenerate.

Note that only the second one has a head to tail structure the first one is energetically very unfavorable for next neighbours AFM exchange. Anyway opposite chirality would require to flip all spins, or one has to state that one compares two different magnetic structures. It must be explained why this would be the basic magnetic structure within the triangle. This is all unclear.

Reply:

Thank you for the comments. To explain the chirality dependence of XMCD spectra, we have added improved figures of model calculation, in which the individual XMCD contribution of each sublattice is shown. In the case of negative spin chirality, the relative arrangement between spin and orbital depends on each sublattice, e.g., the directions between spin and orbital is perpendicular (90°) for the sublattice A, but 30° for sublattices B and C. Thus, the shape of XMCD spectrum for sublattice A and B (C) is qualitatively different as shown by the magenta and light green (blue) curves in Fig4a of the revised manuscript. On the other hand, in the case of positive spin chirality, the relative configuration between spin and orbital does not depend on the sublattice, i.e., the directions between spin and orbital are always perpendicular for sublattice A, B and C. Thus, the XMCD spectrum for each sublattice has same shape, and compensated each other.

Please note that the magnetic structure with negative and positive chiralities do not coexist, and only the triangle magnetic order with negative spin chirality is formed in real Mn_3Sn . The case of positive spin chirality is a fictitious magnetic structure for comparing the spectral shape. Thus, we can choose the magnetic structure shown in the inset of Fig.4a as a basic unit.

The triangle magnetic structure with negative spin chirality is stabilized by exchange and Dzyaloshinski-Moriya interactions in this material [T. Nagamiya, S. Tomiyoshi Y. Yamaguchi, Solid State Communications, **42** (1982) 385-388].

Related to this point, we have improved Fig.4 a, b, and c in the revised manuscript as shown below.

Also, we would like to show revised figure for positive spin chirality (extended data Fig.2e in the revised manuscript) for comparison.

Also, we have added some descriptions related to this point.

➤ **Lines 234-241 in the revised manuscript.**

" The XMCD spectra of each sublattice is also shown in the figure: the individual XMCD contributions from sublattice A, B, and C are illustrated by magenta, dashed light green, and blue lines. In the calculation for Fig. 4a, the Mn spin moments are perfectly compensated, so that the spin contribution for XMCD is expected to be zero. However, a clear large XMCD signal appears. The appearance of XMCD signal can be explained as follows. In Fig. 4a, the shape and

amplitude of XMCD signals are same each other for sublattice B and C (light green and blue lines), but are different between A and B (C) [magenta and light green (blue) lines]. Therefore, the observable total XMCD contribution remains, and the origin is attributed to the T_z term [32]."

➤ **Lines 278-280 in the revised manuscript.**

" Note that the magnetic structure with positive spin chirality (inset of extended data Fig. 2e) is not formed in real Mn_3Sn , and a fictitious magnetic structure for comparison. "

C) Also the results of the sum rules are very uncommon.

ML is clearly negative and beyond zero for $\theta_B = 15^\circ$. But zero for $\theta_B = 90^\circ$. However, one would assume that ml and sz are coupled. I mean that Sz/Lz ratio is usually a constant. Also being directionally parallel or antiparallel.

Are you claiming that ml is coupled to T_z instead? This would be rather strange, would it not? It needs to be discussed.

Reply:

Thank you for the comment. Related to this point, we have revisited sum rule analysis, and improved analysis procedures. Since the XMCD signal of this materials is rather small (~0.4-0.2% of XAS intensity), some artifact signals could be magnified compared with conventional ferromagnets where the XMCD signal intensity is much larger (~10-20% of XAS signal). Here, we would show comparison between original and improved procedure of sum rule analysis.

In our experiment, the difference of XAS for left and right circularly polarization ($\mu^- - \mu^+$) is taken for both positive and negative magnetic fields. And then, the difference of $\mu^- - \mu^+$ between positive and negative magnetic fields is calculated as a XMCD spectrum.

Below, we will show a typical example of analysis procedure. The upper shows the original, and lower shows the improved procedure. In the original procedure, $\mu^- - \mu^+$ data (left panel in the original procedure shown below) show step-like background between low energy and high energy regions. However, this step-like background is an artificial due to the difference of XAS intensity between left and right circularly polarized x-ray by some reasons since the average of $\mu^- - \mu^+$ should be zero in the high energy limit, and should show inversed shape with

field reversal. Although this offset is mostly compensated by the field reversal, a slight offset at the high energy region will still remain in the final XMCD spectrum, and this causes additional contribution to the integrated intensity of spectrum. This additional contribution in high energy region is directly affect the value of orbital moment (m_L).

On the other hand, in the improved procedure, the XAS spectrum is normalized to have a same value in the high energy region (typically $E > 665$ eV). The resultant $\mu^- - \mu^+$ is shown in the lower panel. In this case the high energy value is mostly zero except for the oscillation behavior, which might be originating from the XAFS oscillations. Also, the shape of $\mu^- - \mu^+$ is mostly inverted by the field reversal. These features in $\mu^- - \mu^+$ are more likely than that obtained from the original procedure. By using the improved XMCD spectrum, the q value of integrated XMCD ($\propto m_L$) is much suppressed.

Original analysis procedure

Improved analysis procedure

We have replaced Fig3a by new figure obtained by improved analysis (please see below). Relatively large errors for $\theta_B = 15^\circ$ is coming from the uncertainty of q value due to the oscillation behavior in the high energy region. In the revised Fig3a, m_L moments for both $\theta_B = 15^\circ$ and 90° are mostly zero for whole field range. We would also note that the value of m_S^{eff} is almost independent of analysis procedure, meaning that negative sign and nearly field-independent behavior of m_S^{eff} is intrinsic. Although the sample number in the revised figure is changed, this is due to our wrong labeling between sample number and data in the original manuscript. We have also corrected this point.

C) Can it be explained by defects or an impurity phase that is exchange biased when AFM sets in? can one exclude that? It would explain the disappearance, the weak signal and the different shape (maybe insulating) as well as the larger ML.

Reply:

Thank you for the comments. If the signal is coming from the exchange-coupled impurity phase, the XMCD is expected to be always opposite (antiparallel) to the magnetization even when the field direction is changed. However, in the present case, the magnetic moment (m_S^{eff}) obtained from the sum rule is positive for $\theta_B = 90^\circ$, i.e., parallel to the magnetization, but negative for $\theta_B = 15^\circ$. And improved sum rule analysis shows m_L for both directions are mostly zero. These facts can exclude the exchange-coupled impurity origin for present XMCD.

Related to this point, we have added a following sentence.

➤ **Lines 131-132 in the revised manuscript:**

" This fact also excludes the exchange-coupled impurity origin of present XMCD, where the signal sign and shape are expected to be always opposite to the magnetization. "

There are some minor and technical comments below:

1) Line 2 injections of spin is misleading, as photons have no spin and also injection is a bit strange, may be transfer would be better. I assume you want to say that there is a magnetic angular momentum transfer from the photon to the sample allowing to probe magnetism.

Reply:

Thank you for the suggestion. We have revised corresponding part as follows.

➤ **Lines 54-56 in the revised manuscript:**

" The coupling between circularly polarized light and magnetic moments has various importance in condensed matter physics since this coupling is a source to induce a spin polarization and detection through the angular momentum transfer. "

2) Line 9 Its not the magnetic moments that are 0 but the magnetization

Reply: Thank you for the comment. We have revised corresponding part as follows.

➤ **Lines 61-63 in the revised manuscript:**

" On the other hand, for antiferromagnetic (AF) materials, in which the magnetization is mostly zero due to their compensated magnetic structures, "

3) Figure 2: It is very difficult to understand all panels of Figure 2 from the info provided in the figures and its caption. It might be helpful to always specify both, θ_{in} and θ_B in all panels. In 2c) After looking at it for some time it, the field direction in the caption might the field sweep direction indicated by the arrows? Can one put all the info in the caption and the figure so it is self-explaining?

Reply:

Thank you for the comments. We have revised Fig. 2 and corresponding caption as follows. Please note that there was wrong labeling between measured sample and data. Thus, sample labels have been also corrected.

➤ **Figure 2 in the revised manuscript:**

➤ **Revised caption of Fig.2:**

" **Fig. 2 | a**, Field-orientation dependence of XMCD spectrum for $B = 1$ T (red and blue curve), and XMCD for $\theta_B = 90^\circ$ and $B = 0$ T (dashed sky-blue). For $\theta_B = 15^\circ$ ($\theta_{in} = 25^\circ$), the spectral shape is almost field independent, and the residual XMCD signal is still observed at 0 T as shown in the inset. To measure the XMCD signal at 0 T, 0.1 T is firstly applied, and the field is reduced to be 0 T. **b**, θ_{in} dependence of XMCD peak intensity for L_3 edge and $B = 0.1$ T and 1 T. Corresponding field angle θ_B is also presented in the upper horizontal axis. The dashed line represents a fit proportional to $\cos\theta_{in}$ for the data. **c**, Field-swept XMCD at constant photon energies of 638.8 eV and 640.0 eV for $\theta_B = 15^\circ$ ($\theta_{in} = 25^\circ$, red and magenta curves) and $\theta_B = 90^\circ$ ($\theta_{in} = 100^\circ$, blue and sky-blue curves), respectively. The photon energy for each field direction corresponds to the maximum XMCD intensity at the L_3 edge as indicated by red and blue solid

arrows in Fig. 2a, respectively. Red (blue) and magenta (sky-blue) curves correspond to reducing and increasing field sweep processes, respectively. Field sweep directions are also shown by arrows with the same colors. A large ferroic response with a small switching field of ~ 0.01 T (see inset) is observed for $\theta_B = 15^\circ$ ($\theta_{in} = 25^\circ$), whereas a slight negative slope is observed for $\theta_B = 90^\circ$ ($B \parallel [0001]$ and $\theta_{in} = 100^\circ$). "

4) It is written a large ferroic response is observed (fig. 2 caption), but I do not see any ferroic response, there is no opening of the hysteresis which is required for a ferroic response. It's not compatible with the magnetization data.

Reply: Thank you for the comments.

As shown in the inset of Fig.2c in the revised manuscript, there is a hysteresis with a small switching field of ~ 0.01 T. This switching field is rather smaller than that observed in magnetization measurement (~ 0.04 T) as reviewer suggested, but still reasonably consistent with the range of sample-dependent switching field of this material ($\sim 0.01 - 0.03$ T) as previously reported in refs [1, 3, 6] [Nakatsuji, S., Kiyohara, N., & Higo, T., *Nature* **527**, 212–215 (2015), Ikhlas, M., *et al.*, *Nature Phys* **13**, 1085–1090 (2017), and Higo, T., *et al.*, *Nature Photon* **12**, 73–78 (2018)], which might be originating from some extrinsic origin, e.g., domain wall pinning by impurities and/or defects.

We have added description related to this point.

➤ **lines 145-149 in the revised manuscript:**

" Although the switching field observed in XMCD (~ 0.01 T) is smaller than that of static magnetization measurement (~ 0.04 T, see extended data Fig.1a) with distinct sample. The discrepancy between XMCD and magnetization might be originating from sample dependence with extrinsic origin. The variation of switching field is indeed observed in previous reports [1, 3, 6]."

5) Why are the sample #1, #2, #3, are they different in any way? Is the signal time dependent or any reason why one uses different samples? One should comment on that.

Reply:

Thank you for the comment. The experimental data shown here have been performed in distinct beam time and also different facilities. Also, the XMCD signal of this material is reduced by surface oxidation, so we have used different samples for different experiments.

6) Text to Fig. 2 (remanent component) please describe field history of the zero field measurement then. If there is no field history, don't call it remanent. It might also be helpful to add also in the inset the 0 T data.

Reply:

Thank you for the comment. The history to measure the remanent component is 0.1 T \rightarrow 0 T. We have added following description in the caption of Fig.2a.

➤ **Added sentence in the caption for Fig. 2a in the revised manuscript (Lines 568-569).**

"To measure the XMCD signal at 0 T, 0.1 T is firstly applied, and the field is reduced to be 0 T. "

7) There is a sentence ... "see the sum rule analysis shown in bellow"
may should be "shown below"?

Reply:

Thank you for the careful reading. We have revised corresponding sentence as the reviewer suggested.

8) It might be helpful to label the two different XMCD responses (e.g. different B vs crystal axis and spectral shapes differently)

Reply:

Thank you for the comment. Based on the reviewer's suggestion, we have reconsidered the data presentation. In Fig.2a in the revised manuscript, we have replaced XMCD signal for B=0T and $\theta_B = 15^\circ$ by the data for B=1T to clearly show the anisotropic XMCD response at 1T. We are considering that this behavior is one of the most important features observed in this study.

9) Why is the theta 15 degree not a sum of an XMCD of $\langle s \rangle$ and $\langle Tz \rangle$? , why not visible at theta 15? (How does the difference plot between 1T and 0T, looks like for theta 15 degrees?), as the magnetization is similar for both directions at 1 T.

Reply:

Thank you for the comment. We are considering that the XMCD for $\theta_B = 15^\circ$ is also a sum of $\langle s \rangle$ and $\langle Tz \rangle$ component. The $\langle s \rangle$ contribution for $\theta_B = 15^\circ$ can be typically seen in the Fig.2c, i.e., field swept data. In this figure, the XMCD for $\theta_B = 15^\circ$ has a slightly negative slope at high magnetic field region, and this negative slope is coming from the $\langle s \rangle$ contribution since the $\langle s \rangle$ contribution should be observed as a negative slope. This is also supported that the high-field negative slope for $\theta_B = 15^\circ$ is similar to that for $\theta_B = 90^\circ$. In the modified Fig.2c shown below, we would explain $\langle Tz \rangle$ and $\langle s \rangle$ contributions in the field swept data. In principle, the $\langle s \rangle$ component should be visible in raw spectral shape and sum rule, but hard to see due to the relatively large experimental error by intensity scattering due to the weak XMCD signal in this material.

Related to this point, we have added following descriptions.

➤ **Lines 143-145 in the revised manuscript:**

" Note that the slightly negative slope at high field region is also observed for $\theta_B = 15^\circ$, meaning that the XMCD response for $\theta_B = 15^\circ$ is a sum of ferrioc component and paramagnetic contribution. "

10) Also Sentence: "This highly anisotropic behavior of XMCD is more clearly confirmed in the

field swept data at a constant photon energy" anisotropic in what? Connected to the previous point. Improve formulation

Reply:

Thank you for the comment, and sorry for confusing. The meaning of anisotropy is anisotropy in XMCD intensity. The main point presented here is large intensity variation of XMCD depending on the field directions, while the magnetization for both parallel and perpendicular directions are mostly identical at 1T.

➤ **Lines 137-138 in the revised manuscript:**

" This highly anisotropic response of XMCD intensity is more clearly confirmed in the field swept data at a constant photon energy. "

11) I don't understand the connection between:

"XMCD mostly vanishes at $T = 200$ K." and "the XMCD signal detect the bulk properties."

Reply:

Thank you for the comment. We have revised descriptions related to this point as shown below.

➤ **Lines 211-214 in the revised manuscript.**

" This means that the XMCD signal detect the bulk properties (which is a good evidence to show the bulk origin of present XMCD) since temperature dependence of XMCD would be decoupled from the bulk properties if the present XMCD is arising from the independent surface magnetism from the bulk. "

12) There is an inconsistency between the calculation and the data. Though the overall shape seems in reasonable agreement with the data, the maxima in the calculation in XMCD is coming on the maximum of the absorption spectra, on not on the inflection point. This means that these are clearly are not the same states or they have different energies. In other words the XMCD L3 spectra looks more like a derivative of the L3 spectra than whereas the calculation does not. Of course, these calculations are not very precise, but then they are not such a strong evidence that of the origin of the signal.

Reply:

Thank you for the comment. In the revised manuscript, we have deleted the description to stress the lower-energy peak at the L_3 edge although we are considering that this is one of the characteristic features of present XMCD signal. Instead of this point, we have emphasized nearly field-independent intensity, different sign and spectral shape between parallel and perpendicular field (and also x-ray) directions. To support these experimental results, we have newly added spectral model calculations with small in-plane and out-of-plane magnetizations (Figs. 4b and c in the revised manuscript). From these model calculations, we have confirmed that the spectral shape mostly does not depend on the small canted magnetization, which correspond to the parallel applied field of ~ 1 T. Also, the spectral shape for parallel magnetic field (Figs. 4a and b) is different from that for perpendicular field (Fig.4c). These qualitative agreements between experiment and calculation make our scenario more solidly, and thus the model calculation is a good evidence to reveal the origin of present XMCD.

For the peak position of XMCD, our present model calculation cannot reproduce the low-energy peak observed in experiment as the reviewer pointed out. However, recent our theoretical investigation [ref 32 in the revised manuscript, N. Sasabe, M. Kimata, T. Nakamura, Phys. Rev. Lett. **126**, 157402 (2021)] suggests that the peak position of T_z mediated XMCD can depend on the orbital configurations of material. In this manuscript, we have assumed that the x^2-y^2 type orbital has the lowest energy with localized atomic picture for the model calculation, but this situation might be too simplified due to the itinerant nature of this material. Indeed, the first principles calculation suggests that the lowest orbital of this material consists of a mixture of x^2-y^2 and r^2-z^2 type orbitals with the ratio of $\sim 8:2$. Therefore, we are considering that the situation of our present model calculation is reasonably correct as the first approximation, but further improvement is needed for better understanding of spectral shape.

REVIEWERS' COMMENTS

Reviewer #1 (Remarks to the Author):

The authors have made significant improvement to substantiate the main claim. I would like to recommend it for publication in Nature Communications if the authors could take the following points into account.

In the introduction part, the first experimental observation of the AHE in a cubic noncollinear antiferromagnet Mn₃Pt [Nat. Electron. 1, 172 (2018)] shall be included. In addition to Mn₃Sn and Mn₃Ge, the experimental demonstration of the AHE in noncollinear hexagonal antiferromagnetic Mn₃Ga should be included into the introduction part as well [Adv. Mater. 32, 2002300 (2020)].

Reviewer #2 (Remarks to the Author):

The work reported in this manuscript established the relationship between unusual XMCD signal and magnetic octupole order for the chiral antiferromagnet Mn₃Sn through combined experimental and theoretical studies. The revised version of the manuscript has been significantly improved. In particular, the newly added spectral calculations in Fig. 4 reproduced the major features observed in experiments, thus providing strong support for their claim. The issues raised in my first report have been addressed. I now recommend this manuscript for publication in Nature Communications.

Reviewer #3 (Remarks to the Author):

The authors have considerably improved and clarified their article and replied satisfactorily to all referee comments. In particular, they have redone the sum rules and the results make much more sense now.

Though I still think there are some weak points, such as that results differ for different samples, that the samples oxidize with making a double feature more visible at the L₂ edge, where there is a proposed double feature in the XMCD from the T_z, ..., However, I think the authors provide a reasonable interpretation that seems meaningful and consistent. The fact that the interpretation could really help to extend additional pathways of obtaining information on higher rank multipole moments and its ordering schemes (though T_z is of course a dipole with contains only a limited to view on octupoles), I recommend publication of this manuscript in Nature communication.

One tiny comment

Line 171 sentence:

In usual bulk ferromagnets, $\langle T_z \rangle$ is negligibly smaller than $\langle S_z \rangle$.

The sentence mean that T_z is similar then S_z in usual bulk systems, but for high symmetry systems T_z is often rather small. So maybe they want to be more specific.

Dear Reviewers,

Thank you very much for the reviewers' comments. Here, let us reply to reviewers' comments. We believe that the revised manuscript is now worthwhile publishing in Nature Communications.

Best wishes,

Authors

REVIEWERS' COMMENTS

Reviewer #1 (Remarks to the Author):

The authors have made significant improvement to substantiate the main claim. I would like to recommend it for publication in Nature Communications if the authors could take the following points into account.

In the introduction part, the first experimental observation of the AHE in a cubic noncollinear antiferromagnet Mn₃Pt [Nat. Electron. 1, 172 (2018)] shall be included. In addition to Mn₃Sn and Mn₃Ge, the experimental demonstration of the AHE in noncollinear hexagonal antiferromagnetic Mn₃Ga should be included into the introduction part as well [Adv. Mater. 32, 2002300 (2020)].

Reply:

Thank you for reviewer's recommendation for publication.

Based on the reviewer's comment, we have included references suggested by the reviewer, and changed some descriptions in the introduction part.

Lines 64-65 in the revised manuscript:

" Among many kinds of AF materials, the noncolinear antiferromagnets Mn₃X with hexagonal (X = Sn, Ge, Ga) [4, 5, 6, 7] and cubic (X = Pt) [8] structures are attracting much attention due

to their large anomalous Hall effects."

Reviewer #2 (Remarks to the Author):

The work reported in this manuscript established the relationship between unusual XMCD signal and magnetic octuple order for the chiral antiferromagnet Mn₃Sn through combined experimental and theoretical studies. The revised version of the manuscript has been significantly improved. In particular, the newly added spectral calculations in Fig. 4 reproduced the major features observed in experiments, thus providing strong support for their claim. The issues raised in my first report have been addressed. I now recommend this manuscript for publication in Nature Communications.

Reply:

We would like to appreciate reviewer's suggestion for publication.

Reviewer #3 (Remarks to the Author):

The authors have considerably improved and clarified their article and replied satisfactorily to all referee comments. In particular, they have redone the sum rules and the results make much more sense now.

Though I still think there are some weak points, such as that results differ for different samples, that the samples oxidize with making a double feature more visible at the L₂ edge, where there is a proposed double feature in the XMCD from the T_z, ..., However, I think the authors provide a reasonable interpretation that seems meaningful and consistent. The fact that the interpretation could really help to extend additional pathways of obtaining information on higher rank multipole moments and its ordering schemes (though T_z is of course a dipole with contains only a limited to view on octupoles), I recommend publication of this manuscript in Nature communication.

One tiny comment

Line 171 sentence:

In usual bulk ferromagnets, $\langle T_z \rangle$ is negligibly smaller than $\langle S_z \rangle$.

The sentence mean that T_z is similar then S_z in usual bulk systems, but for high symmetry systems T_z is often rather small. So maybe they want to be more specific.

Reply:

We would like to thank reviewer's suggestion for publication.

Based on the reviewer's suggestion, we have changed a corresponding sentence as shown below in the revised version.

Lines 170-171 in the revised manuscript:

"In usual bulk ferromagnets with high symmetry, $\langle T_z \rangle$ is negligibly smaller than $\langle S_z \rangle$."